# Salivary gland organoid culture maintains distinct glandular properties of murine and human major salivary glands

Yeo-Jun Yoon[1,6], Donghyun Kim [1,6], Kwon Yong Tak[2], Seungyeon Hwang [1], Jisun Kim[1], Nam Suk Sim[1], Jae-Min Cho[1], Dojin Choi[1], Youngmi Ji[3], Junho K. Hur[4], Hyunki Kim [5], Jong-Eun Park [2] & Jae-Yol Lim [1✉]

Salivary glands that produce and secrete saliva, which is essential for lubrication, digestion, immunity, and oral homeostasis, consist of diverse cells. The long-term maintenance of diverse salivary gland cells in organoids remains problematic. Here, we establish long-term murine and human salivary gland organoid cultures. Murine and human salivary gland organoids express gland-specific genes and proteins of acinar, myoepithelial, and duct cells, and exhibit gland functions when stimulated with neurotransmitters. Furthermore, human salivary gland organoids are established from isolated basal or luminal cells, retaining their characteristics. Single-cell RNA sequencing also indicates that human salivary gland organoids contain heterogeneous cell types and replicate glandular diversity. Our protocol also enables the generation of tumoroid cultures from benign and malignant salivary gland tumor types, in which tumor-specific gene signatures are well-conserved. In this study, we provide an experimental platform for the exploration of precision medicine in the era of tissue regeneration and anticancer treatment.

[1] Department of Otorhinolaryngology, Yonsei University College of Medicine, Seoul, South Korea. [2] Graduate School of Medical Science and Engineering, Korean Advanced Institute of Science and Technology, Daejeon, South Korea. [3] National Institute of Dental and Craniofacial Research, NIH, Bethesda, MD, USA. [4] Department of Genetics, College of Medicine, Graduate School of Biomedical Science & Engineering, Hanyang University, Seoul, South Korea. [5] Department of Pathology, Yonsei University College of Medicine, Seoul, South Korea. [6] These authors contributed equally: Yeo-Jun Yoon, Donghyun Kim. ✉email: jylimmd@yuhs.ac

Salivary glands consist of three major glands, namely the parotid gland (PG), submandibular gland (SMG), sublingual gland (SLG), and numerous minor salivary glands[1]. These three major salivary glands are responsible for ~90% of saliva production[2]. Acini, ducts, and myoepithelium are essential for producing and secreting saliva, which is mandatory for lubrication, digestion, immunity, and oral homeostasis[3]. In humans, serous acinar cells in the PGs secrete watery saliva, whereas the SLGs are predominantly comprised of mucous acini that produce viscous and mucin-rich saliva; the SMGs contain both serous and mucous acini[4]. The three major salivary glands feature a distinct cellular heterogeneity for the production of different components of saliva but share common phenotypic features of ducts, which consist of luminal and basal cells, and acini surrounded by myoepithelium[5].

Three dimensional (3D) miniorgans, that is, organoids, have been established from many exocrine glands, including lacrimal, sweat, prostate, and mammary glands[6–9]. This organoid technology could provide a significant opportunity for the advancement of precision medicine in the era of regenerative medicine and anticancer treatment[10–12]. Salivary gland organoids could be generated from the salivary gland-derived stem or progenitor cells[13–15], nonsalivary epithelial stem cells, such as dental follicle stem cells[16], or pluripotent stem cells[17]. However, these organoids displayed somewhat irregular patterns of marker expressions, which were quite different from those observed in tissues. To address this issue, researchers tried different combinations of extracellular matrix (ECM), including a hyaluronate hydrogel[13], a decellularized salivary gland scaffold[16], or a 3D-scaffold culture system that promotes cell-to-ECM and cell-to-cell interactions[18]. Other reports also suggested that some niche factors improved salivary gland organoid formation and recapitulation of tissues[19–21], though they remain to be optimized. To overcome the technical obstacles impeding the 3D reconstitution of the structure and functions of salivary glands, it is imperative to understand the plasticity of adult salivary gland stem/progenitor cells and niche signaling from cell-microenvironmental interactions.

Here, we successfully refined the culture conditions and established a long-term culture of adult salivary gland organoids consisting of salivary acinar, ductal, and myoepithelial cells from all types of murine and human major salivary glands. Each obtained organoid phenotypically and functionally recapitulated the heterogeneity of the structure and gland-specific secretory function of the source salivary gland. Furthermore, this organoid culture technique enabled the cultivation of tumoroids from benign and malignant human salivary gland tumors that retained their distinguishable features. Our tissue-mimicking salivary gland organoids might broaden the insights into the orchestrated interactions among salivary gland cells and reduce the gap between in vitro and sophisticated salivary gland organs in vivo.

## Results

**Establishment of murine adult SMG organoids maintaining heterogeneous cell types with glandular secretory function during long-term culture.** Initially, we found that organoids grown in media containing epidermal growth factor (EGF) and high levels of Wnt signaling exhibited a keratinized core, suggesting that cells inside the organoids lost their epithelial characteristics and underwent squamous metaplasia (Supplementary Fig. 1a, b). We also observed that the expression of keratin 5 (KRT5) and KRT7, which label basal and luminal cells, respectively, was not spatially segregated. Moreover, aquaporin 5 (AQP5) and actin alpha 2 (ACTA2), which label acinar and myoepithelial cells, respectively, were scarcely expressed

(Supplementary Fig. 1a), indicating that further optimized conditions are required to maintain cellular heterogeneity during long-term culture.

Therefore, we tested several compounds to reduce keratinization in murine SMG organoids[22] and found that the addition of retinoic acid (RA) reduced squamous metaplasia in salivary gland organoids (Supplementary Fig. 1c). We then investigated whether EGF-family members, including heparin-binding EGF-like growth factor (HB-EGF), neuregulin 1 (NRG1), and transforming growth factor α (TGFα), could replace EGF and Wnt3a-conditioned medium (CM)[23,24]. We found that among these, NRG1, which has been known to induce organoid proliferation in intestines, mammary glands, and prostate[25–27], highly increased the expression of Wnt compared to EGF, HB-EGF, and TGFα (Supplementary Fig. 1d), suggesting the possibility that NRG1 could substitute EGF and Wnt3a-CM by inducing the endogenous expression of Wnt. Furthermore, we supplemented the growth media with an optimized concentration of A83-01 (ALK inhibitor) and Noggin (BMP antagonist) to attenuate acinar-to-ductal transition (Supplementary Fig. 1e), which has been successfully replicated in other exocrine glands[28,29]. When NRG1-based culture media were used for the murine SMG organoids, we found that fibroblast growth factor 1 (FGF1) and FGF7 could fully support organoid growth and expression of acinar-related genes compared with other conditions (Supplementary Fig. 1f, g). Collectively, we established the organoid culture of each type of murine salivary glands using NRG1-based media containing RA and FGFs but not Wnt3a-CM, which we termed growth and expansion media (hereafter GEM) (Fig. 1a). In particular, to verify the importance of the endogenous Wnt signal induced by NRG1, we treated the organoids with a Wnt secretion inhibitor (IWP-2). The growth of organoids treated with IWP-2 was slower than that of untreated organoids (Supplementary Fig. 1h). We noticed that all of the factors used in the GEM were essential for organoid passaging or growth (Fig. 1b and Supplementary Fig. 2a). We also observed that the murine SMG organoids grew, forming a branching and budding structure from single clonal cells without keratinized core at the early (Fig. 1c) stage and after 7 months, in cultured organoids (passage 30, Supplementary Fig. 2b). Throughout the culture, we observed consistent and robust expressions of ductal and pro-acinar markers for up to 8 months in murine SMG organoids (Supplementary Fig. 2c).

Furthermore, the organoids displayed duct-like structures, which consisted of KRT7[+] luminal cells surrounded by KRT5[+] basal cells (Fig. 1d). However, we did not detect the mature acinar cell marker (MIST1, or *Bhlha15* for transcript) or mucin-positive cells in organoids maintained in the GEM. Therefore, we removed Y-27632 (ROCK inhibitor) and added DAPT (Notch inhibitor) into our media, based on the previous literature[9,30], and referred to this culture condition as DAM (differentiation accelerating media) hereafter (Supplementary Table 1). When cultured in the DAM, we specifically observed that organoids displayed more salivary gland-like features, including mucin-positive cells in endbud-like structures (Fig. 1d). Furthermore, we also found luminal cells with tight junctions (TJP1[+] cells) and lumen clearance (CC3[+] apoptotic cells), MIST1[+] acinar cells, and outermost ACTA2[+] myoepithelial cells in the DAM (Fig. 1e). Interestingly, we observed that this differentiation capacity was maintained at passage 30 without chromosomal aberrations (Supplementary Fig. 2d, e).

To explore the gene expression pattern of the organoid in our culture system, we conducted bulk mRNA sequencing using SMG tissues and organoids cultured in the GEM and DAM. Gene expression profiles revealed that organoids were enriched in ductal cells, retaining the broad expression of gland-specific and

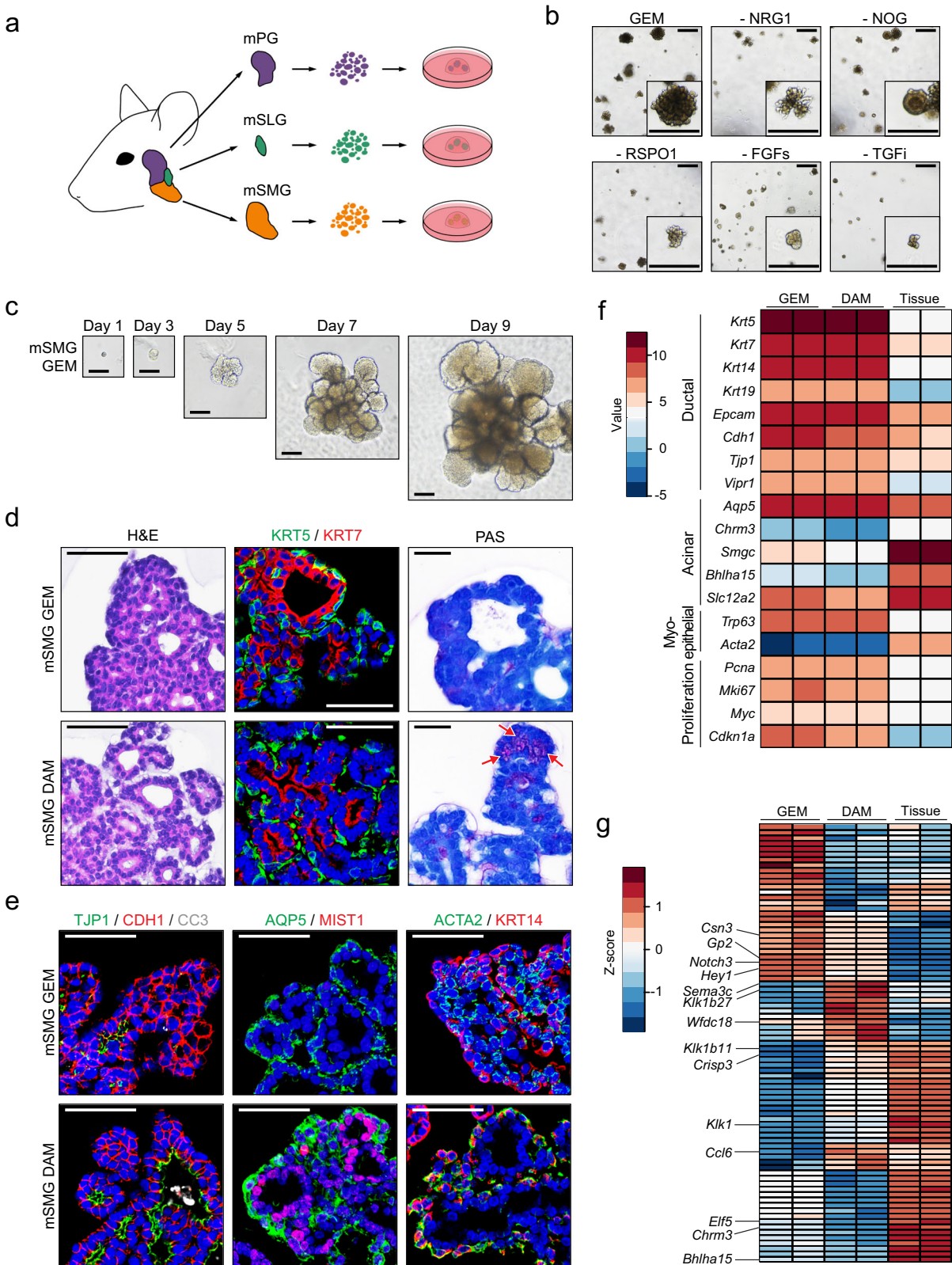

proliferation markers in the GEM and DAM (Fig. 1f). Furthermore, we noticed the upregulation of secretory proteins (*Ccl6* and *Crisp3*) and SMG-specific (*Sema3c, Klk1, Klk1b11,* and *Klk1b27*) genes[31,32], whereas the genes related to the Notch pathway (*Hey1* and *Notch3*) or progenitor (*Elf5, Csn3,* and *Gp2*) (Fig. 1g) were downregulated upon differentiation.

The functions of salivary glands are known to be regulated by neurotransmitters secreted by the parasympathetic or sympathetic nerves[3,33]. To investigate the responses of organoids, we treated the organoids with several neurotransmitters, including carbachol (CCh), isoproterenol (IPR), and vasoactive intestinal peptide (VIP), and found that they displayed swelling

**Fig. 1 Adult submandibular gland (SMG) organoids conserved various salivary glandular cell properties with secretory function in mice. a** Schematic image of experiments using the three major murine salivary gland organoids, including murine parotid gland (mPG), murine sublingual gland (mSLG), and murine submandibular gland (mSMG) organoids. **b** mSMG organoids were maintained in the growth expansion medium (GEM) or GEM lacking each factor, and their growth was observed via brightfield microscopy. Box indicates a representative organoid. Scale bars indicate 500 μm. **c** A mSMG organoid at early passage (p1) was maintained in the GEM and tracked in time-lapse of the images obtained at each time point. Scale bars indicate 100 μm. **d**, **e** mSMG organoids were maintained in the GEM for 9 days or further incubated in a differentiation-accelerating medium (DAM) for another 3 days. **d** mSMG organoids were subjected to hematoxylin & eosin staining (H&E, left), immunofluorescence staining (IF) for ductal markers (middle, KRT5; green, KRT7; red), or periodic acid-Schiff staining (PAS, right). Red arrows indicate PAS-positive regions. Nuclei were stained with hematoxylin (H&E and PAS) or Hoechst 33342 (IF). Scale bars indicate 100 μm. **e** The expression of epithelial (left, TJP1; green, CDH1; red, CC3, cleaved caspase-3; white), acinar (middle, AQP5; green, MIST1; red), and myoepithelial (right, ACTA2; green, KRT14; red) markers was assessed. Nuclei were stained with Hoechst 33342 (blue). Scale bars indicate 50 μm. **f** The expression of salivary gland markers and proliferation-related genes in mSMG organoids cultured under GEM or DAM conditions with SMG tissues ($n = 2$). Heatmap illustrates TMM-normalized expression values. **g** Heatmap of differentially expressed genes (fold change >2, $p < 0.05$) between murine organoids grown under GEM and DAM conditions with SMG tissues ($n = 2$). Heatmap illustrates row z-score of TMM-normalized expression values. All data were representative of three independent experiments.

morphologies upon stimulation (Supplementary Fig. 2f). Furthermore, we detected ATP- or CCh -mediated increases in calcium influx (Supplementary Fig. 2g), which is essential for water secretion[34,35]. Taken together, we successfully maintained the murine SMG organoids harboring cellular heterogeneity, differentiation potential, and functionality during long-term passages.

**Murine adult salivary gland organoids represent distinct glandular structures and retain gland-specific proteins and transcripts.** We next explored whether our organoid culture system could recapitulate the distinct properties of the three major salivary glands. To address this issue, we generated murine PG and SLG organoids employing the same protocol used for the SMG organoids. We observed that the PG organoids had small ducts in a tree-like shape, whereas the SLG organoids had relatively large cystic lumens (Fig. 2a and Supplementary Fig. 3a). As PGs consist of serous acini, whereas SLGs are composed of mucous acini[4], we assessed the presence of mucin-positive cells using periodic acid-Schiff (PAS) staining. Accordingly, we detected PAS-positive cells in the budding structures of the SLG organoids but not in the PG organoids (Fig. 2b). We further examined the expression of amylase 2 (AMY2, a PG-specific marker) and mucin 19 (MUC19, a SLG-specific marker) to specifically distinguish between the PG and SLG organoids[31]. We found that AMY2 was mainly expressed in the PG organoids, whereas MUC19 was observed in the SLG organoids, consistent with the expression patterns in salivary gland tissues (Fig. 2c, d). We further noticed that these organoids retained KRT7[+] luminal cells surrounded by KRT5[+] basal cells, AQP5[+]MIST1[+] acinar cells, and ACTA2[+]KRT14[+] myoepithelial cells when differentiated in the DAM (Supplementary Fig. 3b). Furthermore, ultrastructural images using transmission electron microscopy (TEM) revealed the defined characteristics of serous or mucous granules in acinar cells from the PG or SLG organoids, respectively (Fig. 2e).

Next, we comparatively analyzed the global gene expression profiles in the three major salivary gland organoids. A multi-dimensional scaling plot indicated the clustering of each type of organoid from the PG, SMG, and SLG (Supplementary Fig. 3c). We identified several genes as gland-specific markers in organoids. For example, we detected that *Fam20c*, *Serpine2*, and *Calml3* displayed a PG-specific gene expression in our organoid data, as reported in a tissue database (SGMAP, https://sgmap.nidcr.nih.gov), whereas *Wnt5b*, *Sox2*, and *Dcpp3* were SLG-specific, and *Sema3b*, *Itga8*, *Klk1*, and *Klk1b11* were SMG-specific (Fig. 2f–h). Furthermore, we suggested *Calml3* and *Serpine2* as PG-, *Itga8* as SMG-, and *Sox2* and *Dcpp3* as SLG-specific markers validated in both RNA-seq and qRT-PCR (Supplementary Fig. 4). Gene ontology analysis of these datasets revealed that the genes

linked to the proliferation and function of the salivary gland, including cell-cell adhesion, ion transport, neurotransmitter transport, muscle contraction, and cell chemotaxis, were differentially expressed in the three major murine gland organoids (Supplementary Fig. 3d–h). Collectively, our data indicated that salivary gland organoids retained their tissue-specific features in both the protein and transcript levels.

**Human major salivary gland organoids recapitulate the cellular heterogeneity, structural diversity, and glandular secretory function.** We next explored whether our culture protocol could be adapted to human salivary organoids. To establish human salivary gland organoids, we modified the murine GEM by adding prostaglandin E2 (PGE2), nicotinamide, and GSK3 Inhibitor CHIR99021, which are known to be essential for the maintenance of human organoids derived from other tissues[6,36,37]. Using this approach (Supplementary Table 2), we successfully established organoids from human PG, SMG, or SLG tissues (Fig. 3a), and maintained their growth and expansion for 4 months (Fig. 3b). We found that similar to murine organoids, human salivary gland organoids contained no MIST1[+] acinar cells before differentiation (Supplementary Fig. 5a). Therefore, we tested various differentiation conditions and found that the addition of DAPT combined with the withdrawal of PGE2, nicotinamide, and CHIR99021 from GEM (hereafter human DAM) enabled efficient differentiation of organoids and resulted in the appearance of MIST1[+] acinar and ACTA2[+]KRT14[+] myoepithelial cells along with KRT5[+] basal and KRT7[+] luminal duct cells in all human PG, SMG, and SLG organoids (Fig. 3c and Supplementary Fig. 5b). Furthermore, our organoid culture could maintain expressions of several cell-type-specific markers for up to 4 months (Supplementary Fig. 5c).

We then performed TEM of organoids to observe the ultrastructure of human salivary gland organoids. Our results revealed the existence of various cellular organelles within the secretory epithelial and myoepithelial cells (Supplementary Fig. 6a–d). To evaluate the functionality of human salivary gland organoids, we treated the organoids with ATP or CCh and assessed the calcium influx. We observed increases in intracellular calcium concentration in the human SMG, PG, and SLG organoids (Supplementary Fig. 6e–g), which was conserved through the maintenance of organoids in a dose-dependent manner (Fig. 3d). Furthermore, an increase in calcium influx by CCh was inhibited by atropine, a muscarinic antagonist (Supplementary Fig. 6h). In addition, we detected the swelling of organoids induced by several neurotransmitters, including CCh, IPR, and VIP (Fig. 3e and Supplementary Movie 1–3). We also analyzed the genetic profiles from original tissues and organoids maintained for 3 months to investigate the

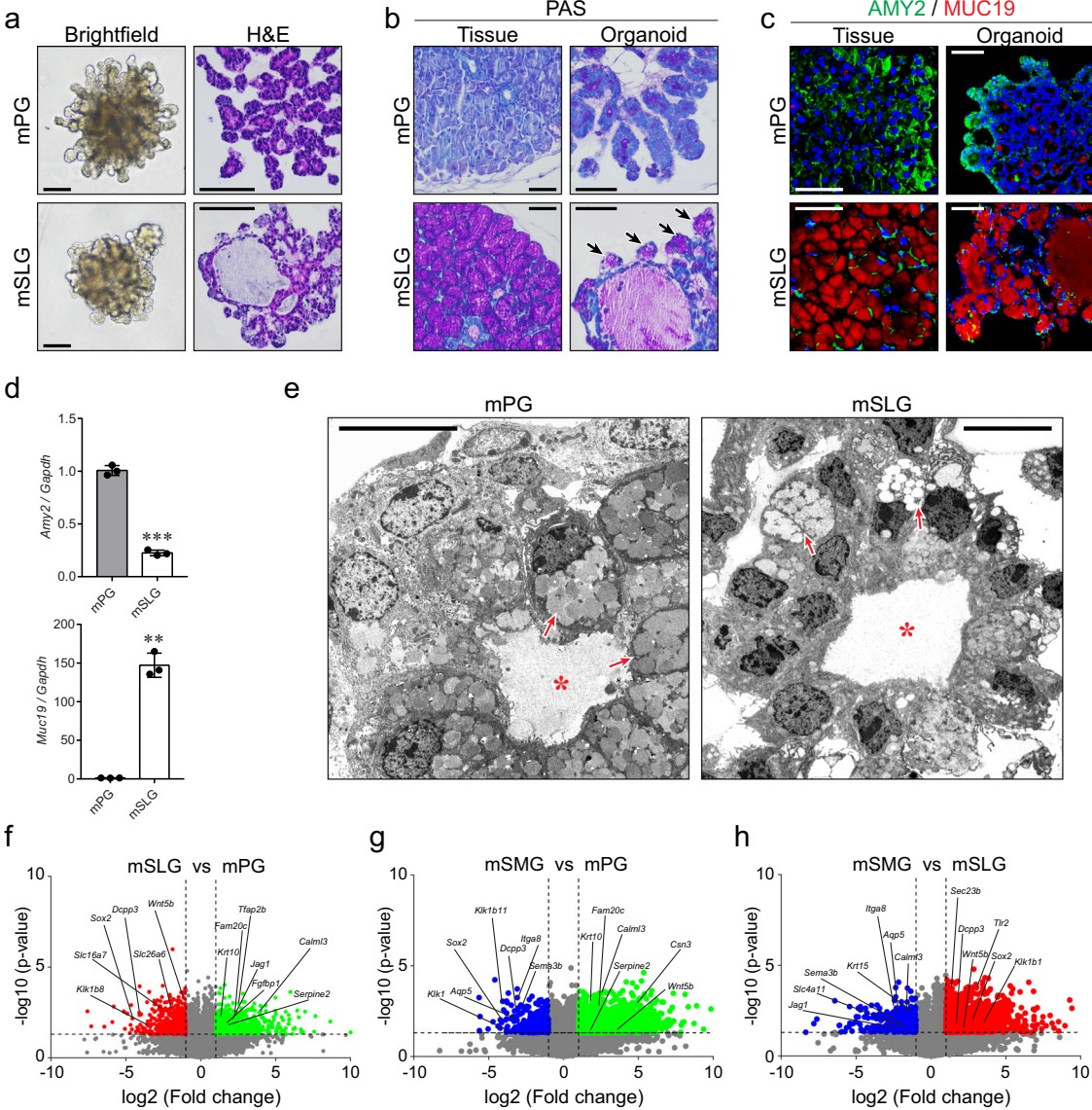

**Fig. 2 Adult salivary gland organoids possessed distinct gland-specific properties in mice. a–d** mPG (top) or mSLG (bottom) organoids were maintained in the GEM for 9 days, followed by 3 days of differentiation in the DAM. **a** Representative images of brightfield microscopy (left) and H&E staining (right). Scale bars indicate 100 μm. **b** The presence of mucin in mPG and mSLG tissues (left) or organoids (right) was assessed via PAS staining. Black arrows indicate the mucin-positive region near end buds. Scale bars indicate 50 μm. **c** IF staining for PG-specific (AMY2, green) and SLG-specific (MUC19, red) markers in tissues (left) or organoids (right). Nuclei were stained with Hoechst 33342 (blue). Scale bars indicate 50 μm. **d** PG-specific *Amy2* or SLG-specific *Muc19* gene expressions were validated via qRT-PCR in organoids. Data were presented as the mean ± SEM ($n = 3$ biologically independent samples). **e** Transmission electron microscopy (TEM) images of mPG (left), and mSLG (right) organoids. Red asterisks indicate internal lumen. Red arrows indicate serous acinar cells or mucous acinar cells in mPG or mSLG organoids, respectively. Scale bars indicate 10 μm. **f–h** Murine salivary gland organoids were maintained in the GEM for 9 days, followed by 3 days of differentiation in the DAM, and subjected to RNAseq analysis ($n = 2$ biologically independent samples). **f** DEGs with a fold change >2 and *p*-value <0.05 between mPG (green) and mSLG (red) organoids depicted using a volcano plot. **g** DEGs with a fold change >2 and *p*-value <0.05 between mPG (green) and mSMG (blue) organoids depicted using a volcano plot. **h** DEGs with a fold change >2 and *p*-value <0.05 between mSLG (red) and mSMG (blue) organoids depicted using a volcano plot. **$p < 0.01$, ***$p < 0.001$. Source Data are provided as a source data file.

accumulation of somatic mutations during organoid development. Using whole-exome sequencing data, we detected a total of 119 and 130 somatic mutations in SMG tissues and organoids, respectively. We observed that 104 variants were conserved among them, whereas 26 somatic mutations were identified as acquired during organoid culture (15 nonsynonymous variants in the coding lesion, five synonymous variants, and six variants in the noncoding lesion). All nonsynonymous variants were considered nonpathogenic (CADD phred score <25), indicating

that our organoid cultures could maintain their genetic stability over time (Supplementary Fig. 6i, j).

Human PG, SMG, and SLG are known to have distinct serous, seromucous, and mucous characteristics, respectively (Fig. 3f, left). To determine whether the organoids recapitulated gland-specific properties, we evaluated the presence of mucin using PAS staining. We accordingly observed the presence of mucin-positive lumen in the SMG and SLG but not in the PG organoids (Fig. 3f, right). Furthermore, we found that mucin 7 (MUC7, a human

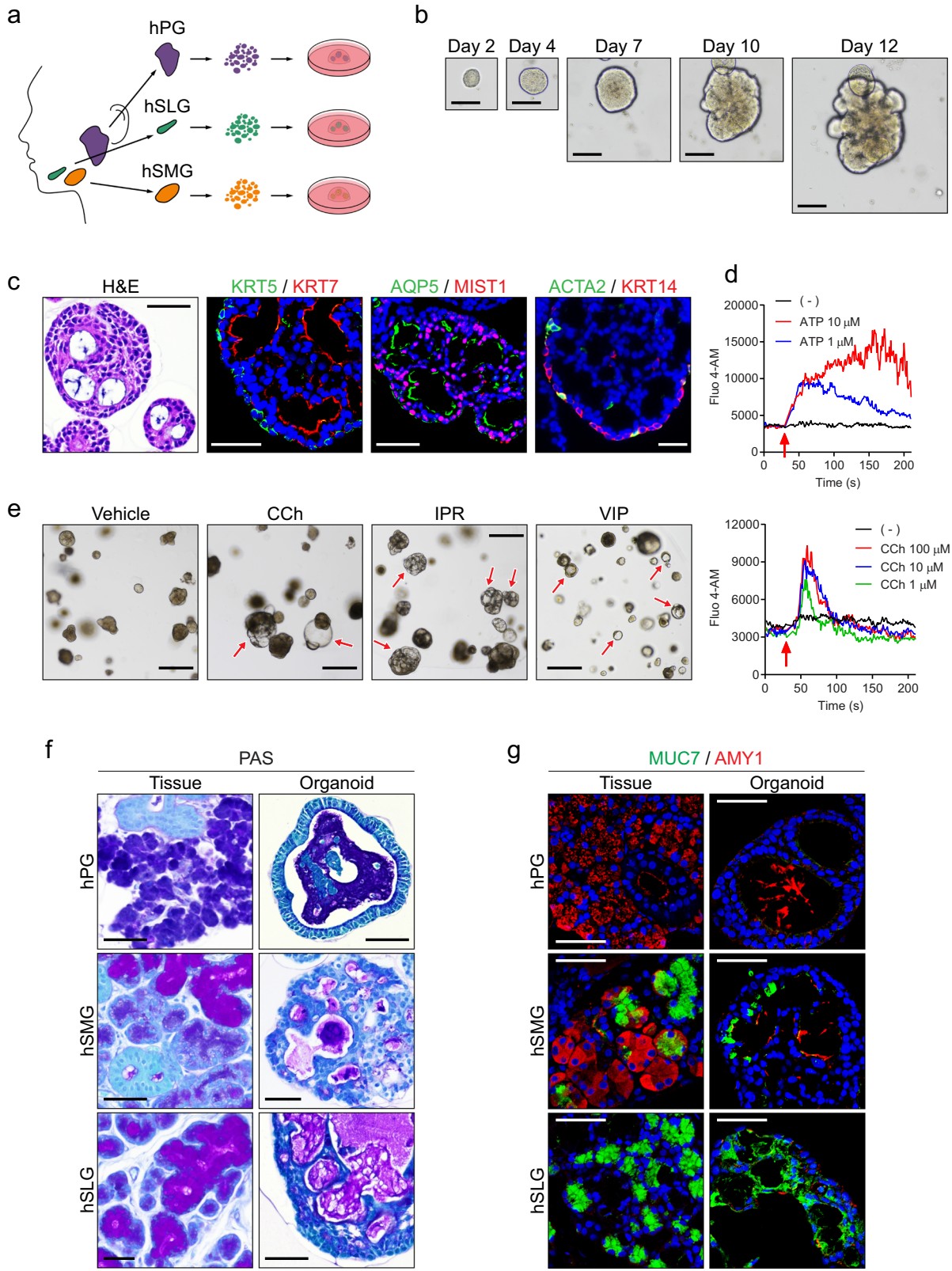

mucous acinar marker) was expressed in the SMG and SLG organoids, whereas, amylase 1 (AMY1, a human serous acinar marker) was exclusively detected in the PG and SMG organoids, consistent with their expression in the source glands (Fig. 3g). Collectively, our results suggested that the human salivary gland organoids displayed cellular heterogeneity, structural diversity, and tissue-specific functionality.

**Human salivary gland organoids derived from single basal and luminal cells showed distinct characteristics**. We observed that human salivary gland organoids exhibited heterogeneous morphologies, consisting of solid- or cystic-organoids, and lacking an obvious branching phenotype (Fig. 4a). More specifically, we found that the human SLG organoids were more cystic, whereas the human PG organoids were mainly composed of

**Fig. 3 Establishment of human salivary gland organoids from PG, SMG, and SLG. a** Schematic image of experiments using three major human salivary gland organoids, including human parotid gland (hPG), sublingual gland (hSLG), and submandibular gland (hSMG) organoids. **b** hSMG organoids were maintained in the GEM. The growth of a single organoid was tracked in the time-lapse of images obtained at each time point. Scale bars indicate 50 μm. **c** hSMG organoids were maintained in the GEM for 2 weeks and incubated in the DAM for another 3 days. Harvested organoids were subjected to H&E staining or IF staining for duct (KRT5; green, KRT7; red), acinar (AQP5; green, MIST1; red), and myoepithelial (ACTA2; green, KRT14; red) markers. Nuclei were stained with Hoechst 33342 (blue). Scale bars indicate 50 μm. **d** hSMG organoids were maintained for 3 months in the GEM, followed by differentiation in the DAM for another 3 days. Single-cell suspensions were prepared, and calcium influx was assessed using Fluo 4-AM in stimulation with either ATP (top) or carbachol (CCh; bottom) at different dosages. Red arrows indicate time points at which cells were treated with stimulants. **e–g** hPG, hSMG, and hSLG organoids were maintained for 1 month in the GEM, followed by differentiation in the DAM for another 3 days. **e** Differentiated hSMG organoids were stimulated with vehicle (DMSO), CCh, isoproterenol (IPR), or vasoactive intestinal peptide (VIP) for 1 h. Then, organoid swelling was observed under a brightfield microscope. Red arrows indicate swollen organoids. Scale bars indicate 500 μm. **f** Sections from hPG (top), hSMG (middle), and hSLG (bottom) tissues (left) and differentiated organoids (right) were subjected to PAS staining. Nuclei were counterstained with hematoxylin. Scale bars indicate 50 μm. **g** hPG, hSMG, and hSLG tissues (left) and differentiated organoids (right) were subjected to IF staining for mucous (MUC7, green) and serous (AMY1, red) acinar cells. Scale bars indicate 50 μm. All data were representative of three independent experiments. Source Data are provided as a Source data file.

dense solid organoids (Fig. 4b). As these morphologic features were consistent with those observed in other human exocrine gland organoids[38,39], we postulated that solid organoids were derived from basal cells, whereas cystic organoids originated from luminal cells. To test this hypothesis, we isolated basal and luminal cells from human salivary gland tissues. There are no known surface markers distinguishing basal and luminal cells in human salivary glands; however, we found that CD49f and CD26, which have been used for the isolation of cells from other glandular tissues[6,38,40], could be applicable for salivary glands (Supplementary Fig. 7a, b). Moreover, the gene expression profile of isolated cells revealed that basal cells were enriched in the CD49f$^+$CD26$^-$ population, whereas luminal/acinar cells were enriched in the CD49f$^+$CD26$^+$ population (Supplementary Fig. 7c). As hypothesized, we detected that the PGs consisted of a larger CD49f$^+$CD26$^-$ population, whereas the CD49f$^+$CD26$^+$ population was enriched in the SLG (Fig. 4c, d).

To investigate organoid formation from isolated cells, we separately maintained organoids from each population in the GEM. As expected, we found that basal cell-derived organoids were solid, whereas luminal cell-derived organoids were cystic (Fig. 4e and Supplementary Fig. 7d). Luminal cells displayed a higher capacity for organoid formation, even in smaller sizes (Fig. 4e, f). However, we did not detect any visible organoid formation in the CD49f$^-$CD26$^-$ population, which contained myoepithelial cells (Fig. 4f and Supplementary Fig. 7c), implying that this culture condition may not be inadequate for maintaining myoepithelial cells. Notably, NRG1 has been proposed as a growth factor supporting mammary and prostate luminal cell maintenance[25,38]. Consistently, we noticed that the receptor tyrosine-protein kinase erbB-3 (ERBB3) and ERBB4 NRG1 receptors were highly expressed in the luminal cell population (Supplementary Fig. 7e). Basal cell-derived organoids contained outer basal cells with inner luminal cells, and KRT14$^+$ basal cells showed the capacity for myoepithelial differentiation upon DAM condition (Fig. 4g and Supplementary Fig. 7f). However, luminal cell-derived organoids consisted of KRT7$^+$ or AQP5$^+$ cells, and AQP5$^+$ cells were capable of differentiating into MIST1$^+$ cells upon DAM condition (Fig. 4g and Supplementary Fig. 7f). Intriguingly, we noticed that ACTA2 was also expressed in luminal organoids, implying that luminal progenitors may have multipotency to be capable of differentiating into acinar and myoepithelial cells[41]. However, luminal cells might acquire plasticity during organoid culture as it has been considered a regeneration model. Furthermore, we observed KRT5$^+$KRT7$^+$ intermediate cells in luminal cell-derived organoids upon the DAM, indicating luminal to basal cell transition, which was suggested in luminal progenitors in mammary glands[42].

Collectively, these results suggested that both luminal/basal progenitors exist in human salivary glands, and our culture system can successfully maintain their growth and distinct characteristics.

**Single-cell RNA sequencing revealed diverse cellular composition and tissue-specific features in organoids.** To analyze the organoids and compare them with the human gland tissues at a single-cell resolution, we performed single-cell RNA sequencing (scRNA-seq) on the organoid and tissue samples. Tissue samples consisted of cells from PG, SMG, and SLG biopsies from different patients, while organoid samples were collected from cultured organoids derived from each biopsy sample. We profiled the cells using the droplet-based scRNA-seq, demultiplexed based on patient genotype, and read a total of 10,287 organoid and 1234 tissue cells. Regarding the organoid data, cells were separated into eight different clusters, as visualized using the uniform manifold approximation and projection (UMAP) embedding (Fig. 5a and Supplementary Fig. 8a). The tissue-specific scRNA-seq data revealed that tissues consisted of duct, acinar, and myoepithelial cells, along with nonepithelial cells including endothelial cells, fibroblasts, and immune cells (Supplementary Fig. 8b).

Analysis of our organoid data revealed the presence of various epithelial populations with cluster-specific features (Supplementary Fig. 8c, d). Expression analysis using established markers revealed high expression of basal (BNC1 and KRT5) and progenitor (KRT15) markers in clusters 1 and 2, with cluster 2 further displaying the high expression of cycling genes (e.g., CDK1, CCNB2, and CDCA2) (Fig. 5b and Supplementary Fig. 8e, f). However, the expression of acinar cell markers (AQP5, BPIFA2, and HTN1) was limited to clusters 3 and 4. We found that differentially expressed genes (DEGs) from both clusters 3 and 4 were highly expressed in the acinar or luminal ductal tissue cells, with some of the genes being particularly exclusive (Fig. 5c, d), implying the acinar-like characteristics of both clusters. This was also supported by diffusion pseudotime analysis, applied for the comprehension of the differentiation trajectory, in which clusters 3, 4, and 5 were likely to be more differentiated (Fig. 5e). In particular, we found that cluster 5 exhibited duct-like characteristics, as its DEGs were slightly enriched in small KLK1$^+$ duct (SERPINB2, SMIM5, and GDPD3) and luminal duct (C15ORF48) populations in tissues (Supplementary Fig. 8g). Clusters 6 and 7, located between the basal and acinar-like clusters in both UMAP and pseudotime trajectory projection, displayed transitional characteristics associated with overlapping expression from other clusters and indistinct DEG patterns, with cluster 6 generally exhibiting a gene signature of myoepithelial cells in tissues (Supplementary Fig. 8h). We also detected the sparsely distributed

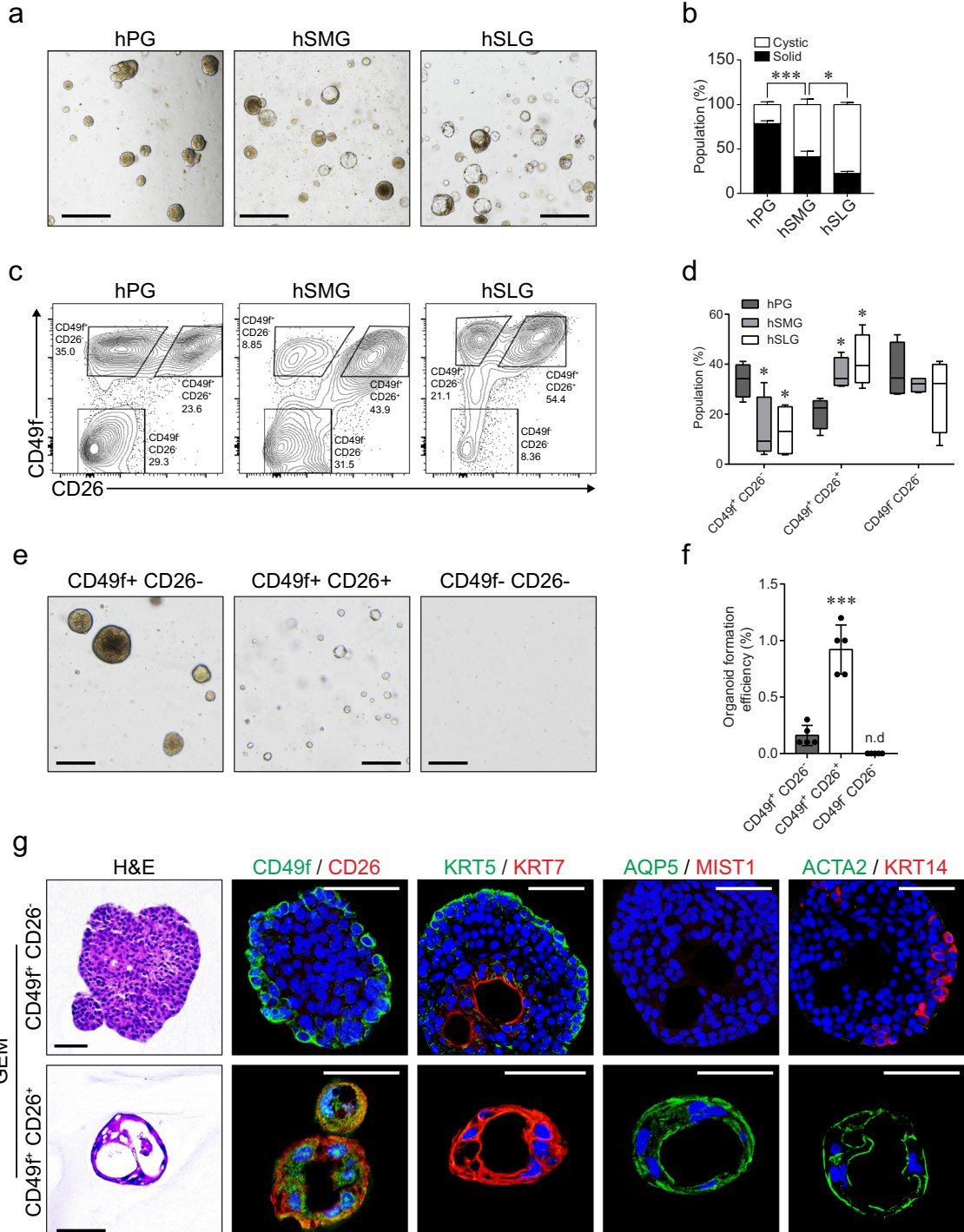

**Fig. 4 Human luminal or basal progenitor-derived organoids maintained their growth and distinct characteristics. a** Brightfield images of human PG, SMG, and SLG organoids cultured for at least 1 month in the GEM. Scale bars indicate 500 μm. **b** Organoids with solid or cystic morphology were counted and statistically analyzed (n = 4 biologically independent samples). **c** Single-cell suspensions were prepared from the three major human salivary gland tissues, and then assessed for the expression of CD49f and CD26 markers in epithelial cells via flow cytometry. **d** The percentage of CD49f+ CD26−, CD49f+ CD26+, and CD49f− CD26− populations were measured via flow cytometry and statistically analyzed (n = 4 biologically independent samples). The box and whisker plots indicate mean, SEM, and 5th and 95th percentiles. **e, f** CD49f+ CD26−, CD49f+ CD26+, and CD49f− CD26− cells from human SMG were isolated and cultured in the GEM for 3 weeks. Brightfield images were obtained for the assessment of organoid growth (**e**) and the evaluation of organoid formation efficiency (**f**). Scale bars indicate 200 μm (n = 5 biologically independent samples). **g** Organoids from CD49f+ CD26− and CD49f+ CD26+ cells were maintained in the GEM for 3 weeks and subjected to H&E staining or IF staining for CD49f (green)/CD26 (red), duct (KRT5; green, KRT7; red), acinar (AQP5; green, MIST1; red), and myoepithelial (ACTA2; green, KRT14; red) markers. Nuclei were stained with Hoechst 33342 (blue). Scale bars indicate 50 μm. Data are representative of at least three independent experiments and presented as mean ± SEM. *p < 0.05, ***p < 0.001, n.d, not detected. Source data are provided as a source data file.

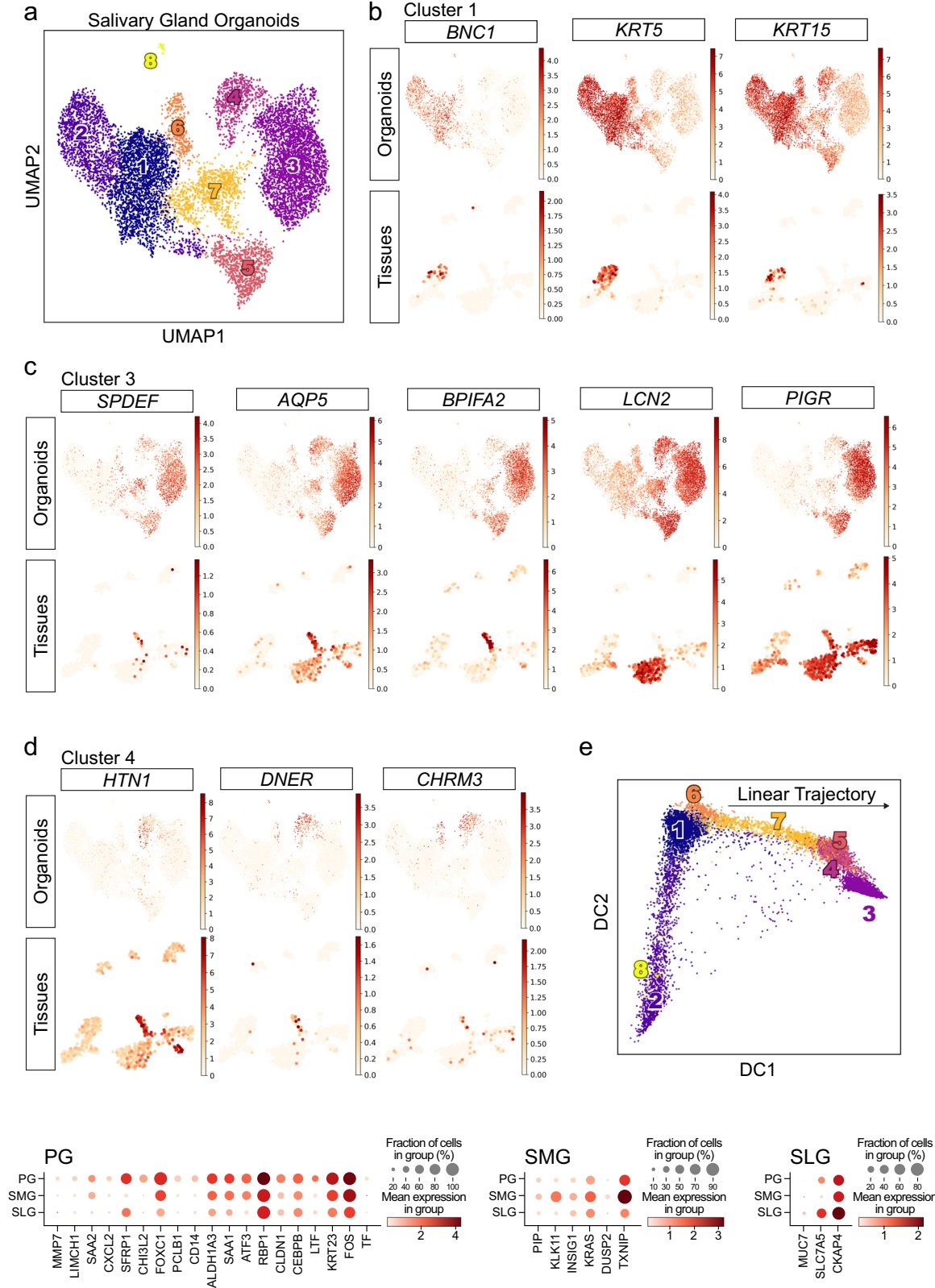

expression of KLK1+ duct (*KLK1* and *SLC26A9*) or myoepithelial cell (*ACTA2* and *MYH11*) gene signatures in organoids (Supplementary Fig. 8i, j), suggesting that these populations were not fully differentiated in our organoid system.

Next, to investigate that our defined clusters were well-conserved during long-term culture, organoids maintained for 3 months were subjected to scRNA-seq and comparatively

analyzed with data from organoids cultured for 1 month. In the UMAP projection of the combined data, the long-term cultured data overlapped well with the original data while maintaining the general structure of original clusters (Supplementary Fig. 9), except small cluster 8 in which some mesenchymal features were observed (Supplementary Fig. 8d), indicating that a few mesenchymal cells remain during short-term culture but are

**Fig. 5 Single-cell RNA-seq revealed cellular heterogeneity and glandular diversity in human adult salivary gland organoids. a–e** Human PG, SMG, and SLG organoids cultured for 1 month in the GEM, followed by differentiation in the DAM for another 3 days, and subjected to single-cell RNA sequencing. Data from three gland organoids were analyzed and shown together. **a** Cell clusters were visualized as uniform manifold approximation and projection (UMAP) for human salivary gland organoids ($n = 10,287$). **b** The expression of cluster 1 (basal)-specific genes (*BNC1*, *KRT5*, and *KRT15*) was visualized in UMAPs for salivary gland organoids (top) and tissues (bottom). **c** Acinar cell (*SPDEF*, *AQP5*, and *BPIFA2*) and luminal duct cell (*LCN2* and *PIGR*) markers were enriched in cluster 3, as visualized in UMAPs for salivary gland organoids (top) and tissues (bottom). **d** Acinar cell markers (*HTN1*, *DNER*, *CHRM3*) were enriched in cluster 4, as visualized in UMAPs for salivary gland organoids (top) and tissues (bottom). **e** Pseudotime analysis displayed inferred divergent trajectory into cycling (downward to cluster 2) or differentiation (rightward to cluster 3) stages from cluster 1 (basal) in organoids. **f** DEGs of PG (left), SMG (middle), or SLG (right) from both organoids and tissues displayed based on scRNA-seq data. The size of the circle represents the percentage of a cell population, while its color depicts gene expression.

removed, and epithelial features are reinforced in long-term culture.

Finally, we identified several DEGs from both organoids (PG, SMG, and SLG) and tissues (Fig. 5f). Surprisingly, regarding secretory proteins[32], we found that PG highly expressed lactotransferrin (*LTF*), whereas prolactin-induced protein (*PIP*) or *MUC7* were mainly expressed in the SMG or SLG, respectively. Collectively, our scRNA-seq data revealed various cell clusters both in tissues and organoids, in which some basal, luminal, acinar, or tissue-specific markers were well-conserved between the tissues and organoids.

**Salivary gland tumoroids recapitulate tumor type-specific characteristics.** Salivary gland neoplasms are heterogeneous diseases composed of various histological subtypes; hence, the clinical behavior of salivary gland cancers can differ according to tumor subtypes and histological grades[43,44]. To address whether our culture system could recapitulate the biologic behavior of different tumor types, we aimed to establish a salivary gland tumoroid culture protocol. We first assessed the possibility of human GEM supporting the growth of organoids derived from benign or malignant tumors. We observed that tumoroids derived from pleomorphic adenoma (PA) sustained their growth in the human GEM, whereas tumoroids from adenoid cystic carcinoma (AdCC) or mucoepidermoid carcinoma (MEC), which are prevalent malignant tumors in salivary glands[45], did not. We solved this issue by reducing the level of A83-01, a potent ALK inhibitor, in our cultures (Supplementary Table 2). In contrast to their paired normal organoids, tumoroids exhibited distinct morphologies (Fig. 6a). Furthermore, hematoxylin & eosin (H&E) staining revealed that each tumoroid originating from different tumors displayed a different cell composition; for instance, squamous cells were typically found in the MEC (Fig. 6b). Next, we assessed the expression of ductal, acinar, and myoepithelial cell markers and found similar expression patterns between tumoroids and their source tissues (Supplementary Fig. 10a–c).

To validate the tumor-specific features of each tumoroid, we assessed the expression of PLAG1, which are representative diagnostic markers of PA[46]. We observed specific expression of PLAG1 in PA tumoroids (Fig. 6c). Moreover, we could detect the expression of c-Kit[47] and MUC1[48] in AdCC and MEC tumoroids, respectively (Fig. 6d, e). Consistently, mRNA expression suggested that tumor-specific features were conserved in tumoroids (Supplementary Fig. 10e). We also noticed that MEC-specific CRTC1-MAML2 fusion transcripts[49,50] were reproduced in the MEC tumoroids (Supplementary Fig. 10d). Next, we addressed the potential usage of salivary gland tumoroids for drug testing. The response of tumoroids to nutlin-3, which inhibits the MDM2-p53 interaction, was used to divide tumoroids into nutlin-3-sensitive or -resistant[37,51] (Fig. 6f, g). Additionally, we employed P53 staining to confirm the accumulation of p53 in the nucleus of nutlin-3-resistant tumoroids, and its lack in nutlin-3-sensitive tumoroids (Fig. 6g).

One of the main treatments against salivary gland tumors is radiation therapy. To explore the possibility of utilizing tumoroids to predict the response of tissues to radiation, we irradiated AdCC or MEC tumoroids using several doses, and accordingly obtained different patterns of response to radiation (Fig. 6h). Especially, we found that one MEC tumoroid (MEC-1), which was derived from intermediate-grade cancers, was radio-resistant. However, we noticed that, in general, tumoroids from low-grade malignancies were radiosensitive, indicating a correlation of radiosensitivity with tumor grade. Therefore, we suggested that our organoid protocol could be adapted to salivary gland tumoroids harboring tumor-specific features, with tumoroids having a potential application in predicting patient-specific responses to drugs or radiation therapy.

## Discussion

Organoid technology has been emerging over the last decade; however, it still requires optimization for the long-time maintenance of epithelial stem/progenitor cells through the use of 3D cultures and the recapitulation of cellular, molecular, and functional traits of the innate tissues of origin[52]. Salivary gland organoids should contain acini and surrounding myoepithelial cells that are essential for their structural and functional recapitulation. We herein introduced an organoid culture for each type of major salivary glands in mice and humans. Our salivary gland organoid culture method maintained the phenotypic features of generated organoids, including their diverse cellular composition, gland-specific marker expression, structure, and function, for a long time (8 months for mouse and 4 months for human organoids). Therefore, our culture method for adult stem cell-based mouse and human salivary gland organoids may facilitate the study of salivary gland regeneration and the pathophysiology of salivary gland diseases and cancers.

There are two major significant differences between our revisited method and the previously reported protocols[14,15,53]. First, EGF and WNT were replaced with NRG1. The expression of EGF and WNT plays a vital role in the normal development of salivary glands[54,55]; however, in the case of salivary gland organoid cultures, EGF and WNT were shown to generate basal cell-rich organoids with a keratinized core. These findings were in line with those observed in prostate organoids[6], which showed that the growth or survival of luminal cells was inhibited at a high EGF concentration and in mammary organoids, in which excessive EGF and WNT in culture conditions led to the excessive keratinization of organoids[27,56,57]. Recently, the addition of exogenous EGF was reported to inhibit the development of proacinar cells in salivary rudiment cultures[58]. As our previous study showed that the growth of mouse salivary organoids was very similar to the embryonic developmental stage of salivary glands, we speculated that the EGF might have also blocked the development of the proacinar cells in organoids[21]. Furthermore, we observed that WNT4 and WNT5b are expressed in ductal lineages of mouse adult salivary gland, while WNT3a is expressed

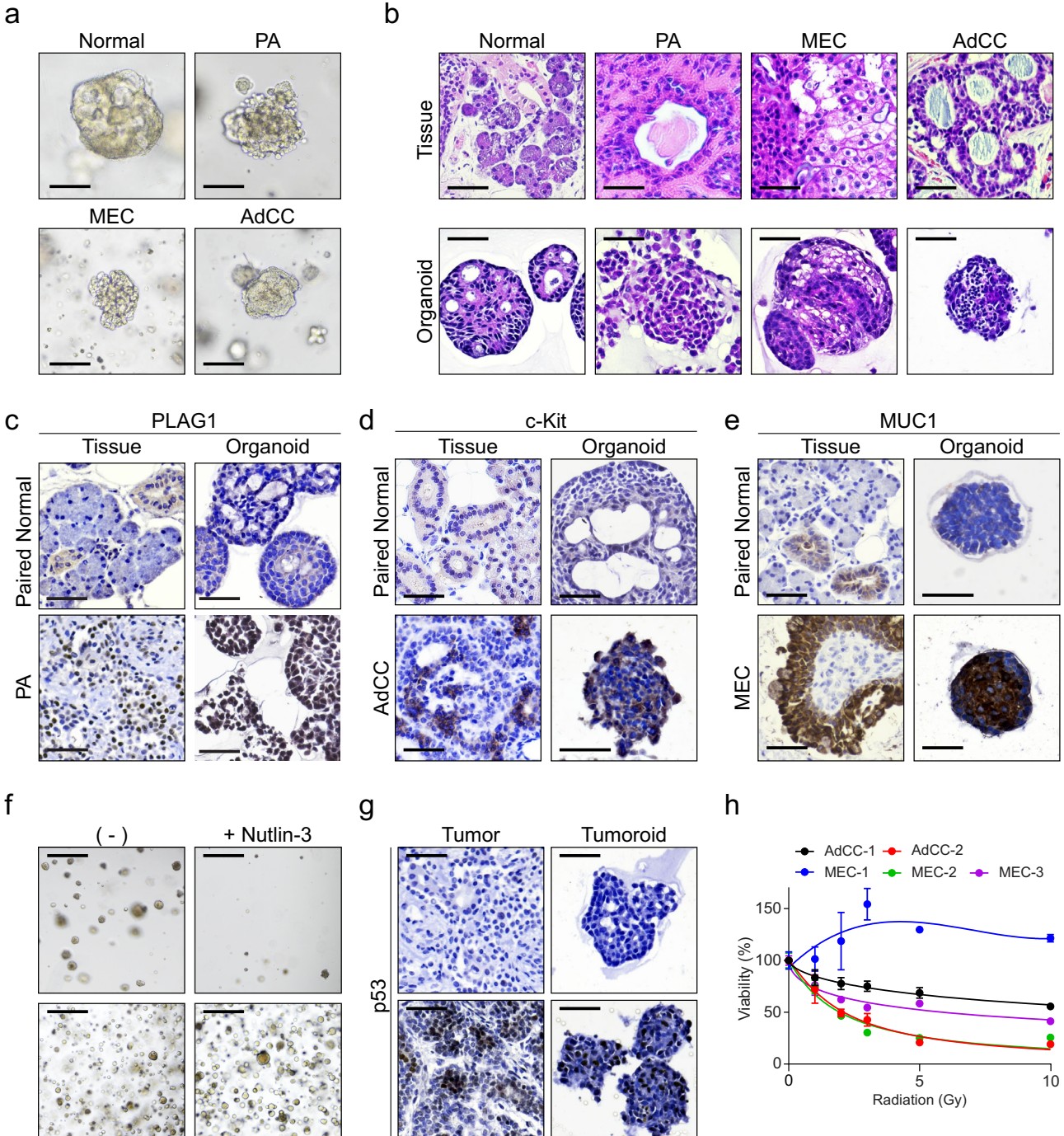

**Fig. 6 Tumor-specific features were conserved in tumoroid models. a–b** Paired normal, pleomorphic adenoma (PA), mucoepidermoid carcinoma (MEC), and adenoid cystic carcinoma (AdCC) tissues were processed for organoid cultures. **a** Organoid growth was observed under a brightfield microscope after 2 weeks of culture in tumor-GEM. Scale bars indicate 100 μm. **b** Histological examination was performed via H&E staining of tissues (top) and organoids (bottom). Scale bars indicate 50 μm. **c** The PA-specific expression of PLAG1 was assessed in tissues (left) and organoids (right) using immunohistochemistry (IHC) staining. Paired adjacent tissues with normal histology were used as control. Nuclei were stained with hematoxylin. Scale bars indicate 50 μm. **d** The AdCC-specific expression of c-Kit was assessed in tissues (left) and organoids (right) using IHC staining. Paired adjacent tissues with normal histology were used as control. Nuclei were stained with hematoxylin. Scale bars indicate 50 μm. **e** The MEC-specific expression of MUC1 was assessed in tissues (left) and organoids (right) using IHC staining. Paired adjacent tissues with normal histology were used as control. Nuclei were stained with hematoxylin. Scale bars indicate 50 μm. **f, g** Tumoroids were cultured with or without the p53 inhibitor nutlin-3 (10 μM) for 5 days. Nutlin-3-sensitive (top) or -resistant (bottom) organoids were assessed for growth under a brightfield microscope **f** while the expression of p53 was evaluated via IHC **g**. Nuclei were stained with hematoxylin in IHC data. Scale bars indicate 500 μm in (**f**) and 50 μm in (**g**). **h** Diverse sensitivity to radiation was observed in MEC or AdCC tumoroids. Cell viability was assessed at 1, 2, 3, 5, and 10 Gy (*n* = 5). Cell viability of tumoroids without irradiation was normalized as 100% viability. Data are representative of at least three independent experiments. Source data are provided as a Source Data file.

in embryonic tissue at scRNA-seq database (https://sgmap.nidcr.nih.gov). Since WNT4 can activate both canonical- and non-canonical WNT signaling, we suggest that multiple WNT ligands have roles in maintaining salivary gland tissue homeostasis and organoid formation. Therefore, a detailed mechanism associated with canonical- and non-canonical WNT signaling remains to be investigated.

Another important point was the greater avoidance of single-cell dissociation during the culture process. The previous report used EpCAM[high] expression to positively select for these cells prior to salisphere formation[15]. However, frequent stress-induced single-cell dissociation may result in the transformation of salivary acinar cells into ductal-like cells, which is similar to that reported in the pancreas[59,60]. This was consistent with a recently published study reporting that partial salivary gland tissue disintegration was good for maintaining acini[61]. We hypothesized that the organoids derived from non-isolated cells could comprise diverse types of progenitors, which support recapitulation of cellular diversity in organoids. Meanwhile, cell-type-specific organoid culture would be beneficial for investigating the role of progenitors in development, differentiation, or regeneration. The potential application of our culture protocol for the generation of cell-type-specific organoids will be explored in future studies.

For the differentiation of salivary gland organoids, we employed the DAPT, a Notch inhibitor, which increased the number of mucous and mature acinar cells (MIST1[+]) in the salivary gland organoids. The function of the Notch pathway in salivary glands has not yet been fully understood, but the activation of the Notch pathway has been shown to promote the activities of stem/progenitor cells in glandular tissues[62–64]. Therefore, we added DAPT to inactivate the Notch pathway and induce the terminal differentiation of each type of stem/progenitor cells. Under the DAM condition, organoids could prevalently differentiate into acinar cells (Figs. 1e and 3c). However, detailed mechanisms of the changes in cell fates according to the activation/inactivation of the Notch pathway require further investigation, as it is unknown whether the events occurring in organoids are similar to the in vivo processes.

Recently, Lin[−]/CD133[+] cells in mouse SMG were suggested as a potential luminal stem/progenitor cell population[65]. However, despite their multipotency in vivo, a sphere-forming assay revealed that these cells were limited to the ductal lineage in vitro. Using our isolated cell-derived human organoids, we maintained the luminal cell-derived organoids in vitro, demonstrating the presence of luminal progenitors. Intriguingly, we observed that luminal organoids gave rise to MIST1[+] acinar and ACTA2[+] myoepithelial cells. Our results suggested that some subpopulations of luminal/acinar cells might undergo trans-differentiation into myoepithelial cells (Fig. 4g and Supplementary Fig. 7f), consistent with the findings in mammary and lacrimal glands[66,67]. Meanwhile, murine organoids exhibited increased branching morphogenesis, whereas human organoids were generally round-shaped. Since murine-derived ECM was used for this experiment, the use of differential ECM suitable for human salivary glands would be beneficial for more sophisticated recapitulation of the salivary glands[68].

The scRNA-seq technology is a powerful tool for the study of cellular diversity on the transcriptomic level compared to the conventional bulk RNA-seq. Although we observed that the organoids consisted of diverse cell clusters while also conserving some tissue-specific features, some limitations remained to be solved. For instance, although we identified acinar-like clusters, these were not fully differentiated in our organoid culture. Especially, several acinar cell markers, including the amylase protein family, were hardly detected in the organoid RNA-seq

experiments. As acinar cells are vulnerable to stress, less stressful conditions during tissue digestion, organoid maintenance, or subculturing would be beneficial for addressing these issues[69,70]. Also, a less harsh and long-term differentiation strategy could allow an improved differentiation of salivary gland organoids, as the conventional differentiation method included a reduction in the Wnt and Notch signaling, which is detrimental to cell proliferation and survival, even in a short period. Nevertheless, our scRNA-seq data supported the idea that our organoid system mainly harbors ductal features, proliferative capacity, and potential for acinar differentiation for long-term periods.

We also modified the NRG1-based culture system and further developed it to grow benign and malignant tumor-derived salivary gland organoids. Tumor organoids recapitulating the genetic variations, morphology, and heterogeneity of the primary tumor could be used in precision medicine[71]. We successfully established tumor organoids from PA, AdCC, and MEC. Tumor organoids maintained the histology of the primary tumor from which they were derived, and retained their unique markers as well as genetic mutations. As diverse tumor subtypes exist and salivary gland tumors are characterized by intratumoral heterogeneity, tumor subtype-specific treatments have been limited[72]. Furthermore, previous 2D cultures enabled the survival of culture-competent cells only during primary cell culture[73,74]. For example, cancer stem cells of salivary AdCC are known as c-kit[+], EGFR[−] cells, but the established cell line includes only EGFR-expressing cells[75,76]. As our generated tumoroids showed diverse cell types, consistent with those in source tumors, they could serve as a suitable platform to investigate the pathophysiology of salivary gland tumors. Furthermore, by using patient-derived tumoroids, we observed differential responses to radiation, suggesting the possibility of employing the use of tumoroids to predict radiation response in a patient-specific manner. Although relatively prevalent salivary gland neoplasms were used for tumoroid cultures, our system could also be applicable for other types of salivary gland cancers. Therefore, it will enable to investigate tumor-specific niche factors or pathways for targeted therapy in diverse tumor subtypes, such as salivary duct carcinoma or malignant mixed tumors.

In summary, we established murine and human salivary gland organoids (Supplementary Fig. 11). The expression of structural markers, expression of tissue-specific markers, functions, and genetic stability were retained through passages. Our data suggested the existence of basal- or luminal progenitors in human salivary glands. Consistently, scRNA-seq also revealed cellular diversity and tissue-specific gene signatures in the human salivary gland organoids. Finally, our data highlighted the potential application of tumoroids for anticancer drug testing and biobanking several types of salivary gland tumors that have not yet been attempted.

## Methods

**Tissue isolation and culture of salivary gland organoids.** Three types of murine major salivary glands (PG, SMG, and SLG) were obtained from 6 to 8-week-old female C57BL/6 mice. Mice were purchased from the Jackson Laboratory (Bar Harbor, ME, USA), fed ad libitum, and maintained at $22 \pm 2\ °C$, $50 \pm 10\%$ relative humidity for 12 h of light-dark cycle (8 a.m.–8 p.m.) under specific pathogen-free conditions in a facility accredited by AAALAC International (#001001). All experiments were approved by the Institutional Animal Care and Use Committee (approval number #2018-0071) at Yonsei University College of Medicine. Human salivary gland specimens were obtained from patients with various diseases, including benign tumors and carcinoma, after acquiring their informed consents and approval from the Institutional Review Board of the Yonsei University Severance Hospital (permission number #2017-0226-001). Population characteristics of patients are as follows. Gender: male (5/17), female (12/17); age: mean 51.0, range 32–79; tumor: benign (6/17), malignant (11/17). Both murine and human salivary gland tissues were cut into small fragments using a razor blade. Fragments were enzymatically digested initially with collagenase type II (#17101015, Thermo Fischer, Waltham, MA, USA) for 1–2 h depending on the tissue size and

subsequently with TrypLE Express (#12604013, Thermo Fischer) for 10 min. Cells were passed through a 70 µm strainer, embedded in growth factor-reduced Matrigel (#356231, Corning, Corning, NY, USA), and supplied with a medium containing relevant growth factors and small molecules.

Mouse GEM contained Advanced DMEM/F12 (#12634010, Thermo Fischer) supplemented with 5 nM NRG1 (#100-03, Peprotech, Cranbury, NJ, USA), 1% homemade RSPO1 CM or 100 ng/mL recombinant RSPO1 (#120-38, Peprotech), 100 ng/mL Noggin (#120-10 C, Peprotech), 5 nM FGF1 (#450-33 A, Peprotech), 1 nM FGF7 (#450-60, Peprotech), 10 µM Y-27632 (#1254, Tocris, Abindon, UK), and 0.5 µM TGFβ inhibitor A83-01 (#2939, Tocris). For the differentiation of murine salivary gland organoids, Y-27632 was removed 7 days after seeding, followed by the addition of the NOTCH inhibitor DAPT (#D5942, Sigma, St. Louis, MO, USA) 3 days after the removal of Y-27632. For the generation of human GEM, murine GEM was modified as follows: FGF1 and FGF7 were replaced with FGF2 (#100-18B, Peprotech) and FGF10 (#100-26, Peprotech), and 10 mM nicotinamide (#N0636, Sigma), 3 µM PGE2 (#2296, Tocris), and 1 µM CHIR99021 (#2520691, Biogems, Colinas, CA, USA) were also added. For the differentiation of human salivary gland organoids, nicotinamide, PGE2, and CHIR99021 were removed and DAPT was added into the differentiation media. For some tumoroids, the concentration of A83-01 was reduced and EGF was added to support the maintenance of organoid growth. Detailed recipes for media composition are described in Supplementary Tables 1 and 2.

Media were changed every 2–3 days, with mouse and human organoid subcultures being conducted every week and biweekly, respectively. The subculture ratio was 1:4 to 1:6 for murine organoids, whereas it was 1:2 to 1:4 for human organoids. For passaging, organoids were harvested using cold ADF12 and dissociated into cell clumps using flame-polished Pasteur pipettes in enzyme-free cell dissociation buffer (#13151014, Thermo Fischer). Dissociated organoids were embedded and incubated in culture media, as described above. For cryopreservation, isolated cells from salivary gland tissues or cultured organoids were resuspended in cell freezing media (CELLBANKER 1, Zenoaq, Fukushima, Japan) supplemented with 10 µM Y-27632 and stored at −80 °C in a freezing container or in liquid nitrogen for long-term storage.

**Quantitative real-time PCR analysis.** Total RNA was extracted using Trizol (#15596018, Thermo Fischer) and reverse transcribed to cDNA using the Prime-Script RT reagent kit (#RR037, Takara, Kusatsu, Japan) according to the manufacturer's protocol. Gene-specific PCR products were measured with a SYBR reporter using the QuantStudio 5 real-time PCR system (Thermo Fischer). Gene expressions were normalized with *GAPDH*. Primers are listed in Supplementary Table 3.

**Organoid proliferation assay.** Cell proliferation was measured using the RealTime-Glo MT (#G9711, Promega, Madison, WI, USA) according to the manufacturer's instructions. Luminescence was detected using a Glomax discover microplate reader (Promega). The results were normalized to the data from organoids cultured in GEM as 100%.

**Image acquisition.** Brightfield, H&E, PAS, and IHC images were obtained using an Eclipse Ti-U2 inverted microscope (Nikon, Tokyo, Japan). IF images were captured using either Eclipse Ti-U2 or the LSM 700 confocal microscope (Carl Zeiss, Oberkochen, Germany). For staining, organoids embedded in Matrigel were incubated with cell recovery solution (#354253, Corning) for 30 min at 4 °C. Then, organoids or tissues were fixed in 4% PFA, embedded in paraffin, and cut into 5-µm-thick sections that were dewaxed and subjected to antigen retrieval for 30 min in Tris-EDTA (pH 9.0). Sections were preincubated with 5% normal serum for 1 h at 25 °C, followed by incubation with primary antibodies overnight at 4 °C. The following primary antibodies were used for IHC or IF experiments; anti-KRT5 (#905904, 1:1000; Biolegend, San Diego, CA, USA), anti-KRT7 (#ab181598, 1:1000; Abcam), anti-ACTA2 (#ab124964, 1:500; Abcam), anti-MIST1 (#ab187978, 1:200; Abcam), anti-AQP5 (#sc-514022, 1:500; Santa Cruz, Dallas, CA, USA), anti-KRT14 (#ab7800, 1:200; Abcam), anti-AMY1 (#sc-46657, 1:200; Santa Cruz), anti-Cleaved Caspase-3 (#9661, 1:500; Cell Signaling, Danvers, MA, USA), anti-TJP1 (#33-9100, 1:50; Invitrogen), anti-CDH1 (#AF648, 1:200; R&D), anti-AMY2 (#ab21156, 1:500; Abcam), anti-MUC19 (#OAEB02736, 1:500; Aviva Systems Biology, San Diego, CA, USA), anti-MUC7 (#orb101843, 1:400; Biorbyt, Cambridge, UK), anti-CD49f (#sc-10730, 1:200; Santa Cruz), anti-CD26 (#TA500733, 1:100; Origene, Rockville, MD, USA), anti-PLAG1 (#H00005324-M02, 1:100; NOVUS, Centennial, CO, USA), anti-c-Kit (#sc-5535, 1:100; Santa Cruz), anti-MUC1 (#14161, 1:200; Cell Signaling), and anti-p53 (#48818, 1:100; Cell Signaling). Biotinylated secondary antibodies and Pierce DAB substrate kit (#34002, Thermo Fischer) were used for IHC. For IF, the following secondary antibodies were used at 1:500 dilution; Donkey anti-Mouse IgG, Alexa Fluor™ Plus 488 (#A32766, Invitrogen), Donkey anti-Rabbit IgG, Alexa Fluor™ Plus 488 (#A32790, Invitrogen), Donkey anti-Mouse IgG, Alexa Fluor™ Plus 555 (#A32773, Invitrogen), Donkey anti-Rabbit IgG, Alexa Fluor™ Plus 555 (#A32794, Invitrogen), Donkey anti-Goat IgG, Alexa Fluor™ Plus 555 (#A32816, Invitrogen), Donkey anti-Rabbit IgG, Alexa Fluor™ Plus 647 (#A32795, Invitrogen), Alexa Fluor 488 AffiniPure Donkey Anti-Chicken IgY (#703545155, Jackson ImmunoResearch, West Grove, PA, USA), and nuclei were

counterstained with Hoechst 33342 (#R37605, Thermo Fischer). The data were analyzed with NIS-elements BR (Nikon) or ZEN (Carl Zeiss) software.

**Functional swelling assay.** Functional swelling of both murine and human submandibular gland organoids was performed by incubating organoids in the DAM. For stimulant treatment, fresh DAM media supplemented with 200 nM VIP (VIPR agonist; #1911, Tocris), 100 µM carbachol (cholinergic agonist; #C4382, Sigma), and 1 µM isoproterenol (adrenergic agonist; #I5627, Sigma) or the same volume of vehicle (DMSO) was added to cultured organoids. For observation and recording, organoids were incubated in live cell imaging solution (#A14291DJ, Thermo Fischer) preheated at 37 °C, and individual organoids were captured and recorded for 1 h immediately after adding the compounds using the Eclipse Ti-U2 microscope.

**Calcium influx assay.** For the detection of the calcium influx, flow cytometry was performed. Briefly, the organoids were incubated with 5 µM Fluo-4 AM (#F14201, Thermo Fischer), 0.02% F-127 (#P2443, Sigma), and 2 mM probenecid (#P8761, Sigma) for 1 h. Then, organoids were harvested and dissociated into single cells using TrypLE for 10 min at 37 °C. Flow cytometry was performed within 1 h after harvesting; carbachol, ATP, or vehicle control were directly added into samples 30 s after the initial baseline recording. In some experiments, 0.1 µM atropine (muscarinic antagonist; #A0257, Sigma) was pretreated 10 min before CCh stimulation. Fluorescence kinetics data were obtained, and Gaussian smoothing was applied to graphs using the FlowJo software (FlowJo, Ashland, OR, USA).

**Karyotyping.** Organoids were incubated in KaryoMAX colcemid solution (#15212012, Gibco) at 0.5 µg/mL for 1 h, followed by single-cell dissociation using TrypLE and filtration via a 20 µm strainer. Pelleted cells were resuspended in hypotonic KCl (75 mM) for 10 min at 37 °C and fixed with a mixture of 3:1 methanol:acetic acid for another 10 min. Cells were mounted on slides and allowed to air dry overnight. Chromosomes were stained with DAPI, and images were obtained using an LSM 700 confocal microscope with a ×63 objective lens.

**Whole-exome sequencing.** Genomic DNA was isolated from human SMG frozen tissues and SMG organoids maintained for 3 months (passage 7) using the MiniBEST universal genomic DNA extraction kit (Takara) according to the manufacturer's protocol. The sequencing library was prepared using the Twist human core exome EF multiplex complete kit (Twist Bioscience, South San Francisco, CA, USA) according to the manufacturer's protocol. After library preparation, all samples were sequenced on a Nextseq (Illumina, San Diego, CA, USA). All sequencing data were preprocessed following the GATK best practice. Analysis-ready bam files were investigated using the Mutect2 caller for somatic mutations and GATK haplotypecaller for germline mutations. All candidates were annotated using the VEP annotator. For accurate analysis, all variants under 5% variant allelic frequency (VAF) were filtered. As all samples were sequenced in an unmatched format, germline variants could not be identified. To calculate conserved somatic variants, candidate variants were evaluated following two filtering criteria; (1) calls showing VAF over 70% were considered germline mutations, (2) variants detected from Mutect2 or Haplotypecaller showing high VAF were considered somatic mutations.

**Flow cytometry analysis and cell sorting.** Human salivary gland tissues were processed into single-cell suspensions as described above and then filtered through a 35 µm nylon mesh (Corning). Cells were resuspended in phosphate-buffered saline supplemented with 0.1% bovine serum albumin, 10 mM HEPES, 25 mM EDTA, and 10 µM Y-27632 to avoid anoikis of epithelial cells. Single-cell suspensions were preincubated with human TruStain FcX (#4322302, Biolegend), and stained with primary antibodies for 30 min at 4 °C. Dead cells were stained with zombie violet (#423114, Biolegend) and gated out. Stained cells were analyzed using LSRFortessa or sorted using FACSAria II (BD, Franklin Lakes, NJ, USA). Data were collected via FACSDiva software (BD) and analyzed using the FlowJo software. The following antibodies were used; anti-CD45-BV605 (#368524, 1:50, Biolegend), anti-CD31-FITC (#303104, 1:20, Biolegend), anti-CD49f-PE-Cy7 (#313621, 1:20, Biolegend), and anti-CD26-PE (#302705, 1:20, Biolegend).

**Bulk RNA sequencing.** Total RNA was isolated as described above, and its concentration was calculated using the Quant-IT RiboGreen (Thermo Fischer). For assessing RNA integrity, samples were run on the TapeStation RNA ScreenTape (Agilent, Santa Clara, CA, USA). Only high-quality RNA preparations, with RIN >7.0, were used for RNA library construction.

A library was independently prepared with 1 µg total RNA for each sample using the Illumina TruSeq stranded mRNA sample prep kit (Illumina). After purification, the mRNA was fragmented and copied into first-strand cDNA using the SuperScript II reverse transcriptase (Thermo Fischer) and random primers, followed by second-strand cDNA synthesis using the DNA polymerase I, RNase H, and dUTPs. These cDNA fragments then underwent an end repair process, an addition of a single "A" base, and ligation of adapters. The obtained products were then purified and PCR-enriched to create the final cDNA library.

Libraries were quantified using the KAPA library quantification kits for Illumina sequencing platforms according to the qPCR quantification protocol guide (Kapa Biosystems, Wilmington, MA, USA) and qualified using the TapeStation D1000 ScreenTape (Agilent). Indexed libraries were then submitted to an Illumina NovaSeq (Illumina) and paired-end ($2 \times 100$ bp) sequencing was performed using Macrogen (Seoul, Korea). Differential expression was analyzed using the DESeq2 and clusterProfiler R Package. The EnrichGO function was used to identify the functions and pathways of differentially regulated genes. A Benjamini and Hochberg adjusted $p$-value $< 0.05$ and absolute log2 fold-change $> 1$ were considered statistically significant.

**Single-cell RNA sequencing**. For the preparation of single-cell suspensions from tissues, cryopreserved cells were thawed. For the preparation of single cells from cultured organoids, organoids were dissociated with TrypLE for 10 min at 37 °C. Live cells were enriched with the dead cell removal kit (Miltenyi Biotec, Bergisch Gladbach, Germany). Cell preparation and dead cell removal were conducted according to the 10× genomics single-cell protocols cell preparation guide and the guidelines for optimal sample preparation flowchart (documents CG00053 and CG000126, respectively).

Libraries were prepared using the chromium controller according to the 10× chromium next GEM single-cell 3'v3.1 protocol (CG000204). Briefly, cell suspensions were diluted in nuclease-free water to achieve a targeted cell count of 10,000. The cell suspension was mixed with the master mix, and loaded with single-cell 3'v3.1 gel beads and partitioning oil into a chromium next GEM chip G. RNA transcripts from single cells were uniquely barcoded and reverse-transcribed within droplets. cDNA molecules were pooled and then subjected to an end repair process, an addition of a single "A" base, and ligation of adapters. The final products were then purified and PCR-enriched to create the final cDNA library. Purified libraries were quantified using qPCR according to the qPCR quantification protocol guide (Kapa Biosystems) and qualified using the Agilent Technologies 4200 TapeStation (Agilent). Libraries were sequenced using the HiSeq platform (Illumina) according to the read length in the user guidelines.

The obtained fastq files were processed using cellranger count (CellRanger 4.0.0, 10× genomics) with a custom human reference genome GRCh38. The processed files were demultiplexed with Souporcell[77]. The demultiplexed output quality was confirmed by comparison with patient genotype data. For both dataset of tissue and organoid, we filtered low quality cells with <1500 measured genes and a high percentage of mitochondrial contamination (>20). After filtering, data in each cell were normalized to log (CPM/100 + 1), and highly variable genes were identified. After scaling the expression levels of these genes, PCA was performed in variable gene space. Next, 50 principal components were used for neighbor computing, and UMAP dimensionality reduction was computed. All steps were performed using functions implemented in the Scanpy package. The batch effect was corrected via Harmony (version 0.0.5), and graph-based clustering at multiple resolutions was performed. Then, we applied the binary marker gene detection algorithm, assuming that different clusters should have several marker genes robustly distinguishing them with binary logic. After that, we selected the optimal resolution that does not lead to the over-clustered segments without specific marker genes[78].

**CRTC1-MAML2 fusion gene detection**. The detection of the CRTC1-MAML2 fusion oncogene in MEC tissues and tumoroids was performed as previously reported. Briefly, RNA was extracted from tissues and tumoroids, and subjected to cDNA synthesis as described above. Initial PCR was performed using the following primers: *CRTC1* 5'-AAG ATC GCG CTG CAC AAT CA-3' and *MAML2* 5'-GGT CGC TTG CTG TTG GCA GG-3'. Consecutively, 0.2 μg of initial PCR products were subjected to nested PCR. The primers used for the detection of the *CRTC1-MAML2* fusion genes were *CRTC1* 5'-GGA GGA GAC GGC CTT CG-3' and *MAML2* 5'-TTG CTG TTG GCA GGA GAT AG-3'. To verify the fusion gene PCR products, sequencing was performed using Macrogen with the primers used in nested PCR, and the sequences were compared with the reference fusion gene sequence (GenBank #AY0324.1).

**Radiation model in tumoroids**. Tumoroids derived from carcinoma (MEC and AdCC) were harvested and subcultured in 48-well plates as described above. Then, 3 days after seeding, the organoids were irradiated using the X-Rad320 (Precision X-Ray, North Branford, CT, USA). Separate culture plates were used for different radiation doses ranging from 0 to 10 Gy. Media were changed after radiation, and organoids were cultured for another 4 days. Cell viability was measured via the evaluation of ATP levels using the CellTiter-Glo 3D reagent (Promega), according to the manufacturer's protocol. The results were normalized to baseline (0%, staurosporin 1 μM) and positive (100%, 0 Gy) controls.

**Statistics and reproducibility**. The data presented in this study were representative of three independent experiments and statistically analyzed using GraphPad Prism (version 7.00, GraphPad Software, San Diego, CA, USA). Unpaired two-tailed Student's $t$-test (two groups) and one-way ANOVA with Tukey's posthoc test (more than two groups) were performed to compare the values and evaluate statistical significance. $P$ values $< 0.05$ were considered significant. The experiments were not randomized, and investigators were not blinded to allocation during experiments and outcome assessment.

**Reporting summary**. Further information on research design is available in the Nature Research Reporting Summary linked to this article.

## Data availability
The RNA sequencing data generated in this study have been deposited in the NCBI's Gene Expression Omnibus database under accession code "GSE184091" for bulk RNA-seq or "GSE184526" for scRNA-seq. All other relevant data supporting the key findings of this study are available within the article and its Supplementary Information files or from the corresponding author upon reasonable request. Source data are provided in this paper.

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

## Acknowledgements

This work was supported by the Bio & Medical Technology Development Program or Basic Science Research Program through the National Research Foundation of Korea (NRF) funded by the Ministry of Science and ICT (NRF-2018R1A2B3004269, NRF-2020M3A9I4039045, NRF-2021M3A9I402444712, and NRF-2021R1C1C101009412, Republic of Korea) and the Ministry of Education (NRF-2019R1I1A1A0106354513 and NRF-2020R1I1A1A01070328, Republic of Korea).

## Author contributions

Y-J.Y., D.K., and J-Y.L. conceived the study. Y-J.Y. carried out the laboratory work related to murine organoids. D.K. carried out the laboratory work related to human organoids. S.H., J.K., and Y.J. carried out staining, imaging, and data acquisition. J-M.C. and D.C. provided materials. Y-J.Y., D.K., K.Y.T., N.S.S., and J-E.P. analyzed the data. Y-J.Y., D.K., K.Y.T., J-E.P., and J-Y.L. wrote the original and revised manuscripts. Y.J., J.K.H., and H.K. participated in manuscript revision. All authors reviewed and approved the final manuscript.

## Competing interests

The authors declare no competing interests.
