## [Peer Review File · Nature Communications]

REVIEWER COMMENTS

Reviewer #1 (Remarks to the Author):

The manuscript by Yoo et al. entitled "Long-term salivary gland organoid culture maintains distinct glandular properties of murine and human major salivary glands" demonstrated 3D culture methods for mouse and human salivary glands. The authors modified the culture method reported previously to reduce keratinization. RA was found to be useful to reduce keratinization. Interestingly, NGR1 was shown to substitute EGF and Wnt3a-condition medium by inducing the endogenous expression of Wnt. NRG-based culture medium with RA, FGF1, and FGF7 was established as growth and expression media (GEM). In addition, differentiation-accelerating media (DMA) was also established based on GEM medium. DMA seems to make it possible to maintain the differentiated organoid at passage 30 without chromosomal aberration. The authors also showed that these culture methods could be applicable for the 3 major salivary glands. Next, they succeeded in establishing culture methods of human major salivary gland organoids by modifying GEM. Indeed, it is of interest to establish these culture methods, but the author did not go into molecular mechanisms related to organoid formation. In addition, the authors used many reagents for SMG organoid culture. There was no description of how the optimal concentration of the reagents was determined. The dose-dependent effect of each reagent should be examined. In addition, the activity of each reagent needs to be evaluated to exclude the possibility of degradation of the reagent. Next, tumor organoids were shown to be established by these culture methods. However, organoid culture methods for salivary gland tumors have been reported (Int. J. Cancer 2021; 148:193-202). The authors should describe the differences between them.

Major points:

1. In Figure S1, keratinization of organoids appeared in the presence of EGF and Wnt3a-condition medium. How much frequent keratinization happens?
2. NGR can substitute EGF and Wnt3 through increasing endogenous Wnt expression. To confirm this possibility, it is necessary to examine whether a decrease of Wnt3a expression using siRNA affects organoid formation in the presence of NGR.
3. In Fig.3D, calcium influx by ATP and CCh was showed. However, both dose-dependency and inhibition by antagonist atropine needed to be examined.
4. On page 12, lines 243-247, the authors mentioned that ACTA2 was also expressed in luminal organoids, implying that luminal progenitors may be a common ancestor of acinar and myoepithelial cells. However, the phenotype of differentiated cells after cell culture is not always recapitulating normal conditions in vivo, because myoepithelial cells in exocrine glands, such as salivary glands and mammary glands, are known to acquire multipotency after cell culture by dedifferentiation.

Reviewer #2 (Remarks to the Author):

In vitro primary salivary gland cell cultures including organoids generally do not exhibit the diversity of parenchymal cell types characteristic of the native glands, and are comprised mainly of duct-like cells that are not fully differentiated. The authors aim to create salivary organoids from adult salivary gland cells that exhibit diverse parenchymal cell types more characteristic of natural salivary glands. Using murine primary cells isolated based on previously published methods for adult submandibular salivary progenitor cell isolation, they use growth factors known to promote exocrine gland development and inhibitors used on other organoid systems to develop a Nrg1 and small molecule supplemented media that together with FGF1 and FGF7 fully supports long term submandibular salivary organoid growth and expansion that they call GEM. In contrast to the media formulations from the previous referenced work, GEM supports ductal cells without squamous metaplasia and with distinct basal and luminal marker expression as well as cells exhibiting limited expression of the acinar marker Aqp5 and the myoepithelial marker Acta2, but does not support acinar markers such as Mist1 or PAS positive mucin expression. They then develop a stepwise differentiation protocol where the GEM is replaced by a differentiation accelerating media (DAM) with key attributes including removal of Y27632 and DAPT, which promotes differentiation of organized groups of cells exhibiting expression both acinar (properly localized Aqp5, Mist1, and PAS) and myoepithelial (basally localized coexpression of Acta2 and K14) markers at the protein level as well as related transcriptional changes. Organoids show some level of neurotransmitter response similar to native glands. These data are a significant advance over previous work with adult cells demonstrating long term expansion with independent expansion and differentiation of the major parenchymal cell types, which they demonstrate is not accompanied by overt chromosomal instability. They establish murine and human parotid, submandibular, and sublingual salivary organoids that exhibit duct, acinar, and myoepithelial cell features and are reminiscent of the respective parental glands. They adapt their methods to create organoids from benign and malignant human salivary tumors, called tumoroids, Human tumor-derived tumoroids were obtained from 3 different tumor types and the tumoroids were shown to recapitulate the original tumor type. Tumoroids were then shown to be amenable to patient-specific in vitro treatment sensitivity assays using small proof of principle assays with nutlin-3 resistant growth tests or in vitro irradiation viability tests, suggestive of future clinical applicability in precision medicine approaches. The manuscript is well written, the data are compelling, and the work is significant with potential future clinical applications.

There are a few points that should be clarified:

Figure 1 and S1. The authors reference Maimets et al. 2016 for the previous culture method upon which the current manuscript salivary organoid cultures are based. Of note, Maimets et al 2016 used EpCAM-high expression to positively select for these cells prior to salsphere formation and the current work methods section does not include this step. This is a significant difference that should be acknowledged in the text.

Figure 4 and S4. Differences in solid vs cystic morphologies of the human salivary organoids were determined to result from whether the organoids derived from basal or luminal cells, characterized by or isolated using CD49f and CD26. It seems that the organoids derived from FACS-isolated basal or luminal cells were used to form organoids and assayed for differentiation only with human GEM, and not with human DAM. This is unfortunate and it would be of great interest to see the isolated cell populations capacity for differentiation under conditions that accelerate differentiation. The authors should explain why DAM was not used or include a comparison for DAM vs GEM media as the comparison will help establish the contributions of each type of media.

Figure 5 and S5. scRNA-seq analysis show that some of the organoid clusters are duct-like, some are acinar-like and several show incomplete differentiation with characteristics of more than one epithelial cell type. Gland specific differences are noted. Could the authors explain in more detail how they came up with the number of clusters that they report and comment on the stability of their clustering?

Although 8 clusters are labeled, the cell identity of each cluster is not made clear in the main figure.

ScRNA Seq has been published previously for adult SMG by Hauser, et al 2020, and they identified more clusters in adult glands in vivo. It would be helpful to know how the organoids relate to the in vivo glands. This could potentially be accomplished by direct comparison with the published data.

Reviewer #3 (Remarks to the Author):

Yoon et al demonstrated that “Long-term salivary gland organoid culture maintains distinct glandular properties of murine and human major salivary glands”. Authors suggested that a modified culture method for the generation of salivary glands organoid was suitable to long-term culture, and maintain the major salivary glands cell types in human and murine organoids. And the authors also presented culture conditions for tumor salivary gland organoids. Their overall findings are interesting, however, there are several issues in the manuscript and the data as presented, which should be addressed as follows.

In this study, the authors emphasized that salivary gland organoids can be cultured for a long time and retain their properties. However, I could not see the comparing controls to describe the characteristics of the long-term culture. And I also could not find the time-dependent experiment, such as day or month, for assessment of long-term culture properties. And also I could not find the quantitative analysis (ex, qRT-PCR analysis) in the main figures.

1. To make the GEM, the authors used NRG1 to replace the Wnt3A condition medium. Because NRG1 treatment raises the expression of various Wnt genes at the endogenous level, the authors assured that NRG1 could be substituted with the addition of Wnt 3a. Is the endogenous levels of Wnts expression sufficient to produce salivary gland organoids? The expression of not only Wnt3a but also Wnt4, Wnt5b, and Wnt10b is increased 10-fold or more by treatment with NRG1. In order to confirm that the endogenous level of Wnt3a concentration is sufficient, downregulation of Wnt3a after NRG1 treatment will be required, and also confirmation of other Wnt parts will be required during the generation of organoids.

2. In figure 1C, D, E. How many days is 30 passage? The authors showed 9 days organoid (mouse) or 12 days organoid (human) in the main figures. Authors should present the change of cell type marker expression (qRT-PCR analysis) and phenotypes by date.

3. Figure 1F and G, authors should perform re-analysis using GEM and DAM RNA seq-result compared to mouse tissue for assessment of organoid in GEM and DAM condition.

4. Figure 2C and E,F,G. Authors should perform qRT-PCR analysis for validation of each expression level. Only ICC data and RNA-seq data are not enough for elucidation of the result.

5. Figure 3. Authors also should present the change of marker expression (qRT-PCR analysis) and phenotypes by date. In this system, the author demonstrated the 3 month culture of organoids. These are important points in this study. After long-term culture, the cell types and specific cell type markers should be maintained. Thus authors should present the qRT PCR results of cell type markers by date or month. Moreover in figure 3D, the authors suggest the result of organoids function by neurotransmitters (ATP, CCh). The Authors also should exhibit the result of calcium influx by date or month. Summarizing these results, I can decide that the new culture system is better than the old system for long-term culture.

6. Figure 5A. It is not clear for performing scRNA-seq in this study. According to the tile, it is assumed that this study established "salivary gland organoid long-term culture." In general, for scRNA-seq data, differentiation of salivary gland organoids needs to show how they have corresponded to the cell types. However, this does not mean the results of long-term maintenance of organoids. This study needs to provide the data to demonstrate the purpose of long-term maintenance.

7. scRNA-seq : This study should compare the well-differentiated salivary gland organoid with long-term maintenance salivary gland organoid in cell-type differences in order to highlight the importance of this protocol. From this data, there is no clue for demonstrating long-term salivary gland organoids are well maintained. Actual salivary gland organoid data of long-term maintenance should be provided.

6. Figure 6. Authors also should present the expression level of cancer markers (qRT-PCR analysis) compared to normal organoid and tissue.

7. Describe the figure legends in more detail.

Reviewer #1 (Remarks to the Author):

The manuscript by Yoon et al. entitled "Long-term salivary gland organoid culture maintains distinct glandular properties of murine and human major salivary glands" demonstrated 3D culture methods for mouse and human salivary glands.

The authors modified the culture method reported previously to reduce keratinization. RA was found to be useful to reduce keratinization. Interestingly, NGR1 was shown to substitute EGF and Wnt3a-condition medium by inducing the endogenous expression of Wnt. NRG-based culture medium with RA, FGF1, and FGF7 was established as growth and expression media (GEM). In addition, differentiation-accelerating media (DAM) was also established based on GEM medium. DAM seems to make it possible to maintain the differentiated organoid at passage 30 without chromosomal aberration. The authors also showed that these culture methods could be applicable for the 3 major salivary glands. Next, they succeeded in establishing culture methods of human major salivary gland organoids by modifying GEM. Indeed, it is of interest to establish these culture methods, but the author did not go into molecular mechanisms related to organoid formation. In addition, the authors used many reagents for SMG organoid culture. There was no description of how the optimal concentration of the reagents was determined. The dose-dependent effect of each reagent should be examined. In addition, the activity of each reagent needs to be evaluated to exclude the possibility of degradation of the reagent. Next, tumor organoids were shown to be established by these culture methods. However, organoid culture methods for salivary gland tumors have been reported (Int. J. Cancer 2021; 148:193-202). The authors should describe the differences between them.

Ans> We appreciate your valuable comments. We performed additional experiments to address your questions, and added more detailed explanations for better understanding. According to your suggestion, we provided the data regarding the dose response of A83-01, Noggin, RSPO1-CM, and NRG1 in murine salivary organoid culture medium. We observed that A83-01 and RSPO1-CM at a higher concentration and Noggin at a lower concentration produced larger organoids (Figure S1E). However, we also found that excessively large organoids resulted in cell death or stress-induced dedifferentiation of cells inside the organoids, as growth factors were compromised to enter the organoids. To maintain organoids in healthy condition, we chose 0.5 μ M A83-01, 100 ng/mL Noggin, and 1% RSPO1-CM in mouse GEM. However, we found no noticeable difference in organoid growth

when a different dose of NRG1 was used (1, 5, and 10 nM, data not shown). Other reagents were used at the dose previously reported.

FGF1 and FGF2 are sensitive to temperature among various supplements in the organoid culture medium ¹. The organoid media were kept at 4°C during storage and used within 1 week after culture media preparation. The culture media aliquoted for single-use were then incubated at 37°C for 10 min just before use. Nonetheless, we observed no significant difference in organoid growth when new media or media incubated at 37°C for 3 days were used for organoid growth (data not shown), indicating that degradation of reagents in our media was minor in terms of organoid proliferation during media change periods.

We speculated that highly activated EGF and WNT signaling could result in ductal lineage enrichment and squamous metaplasia since EGFR expression and WNT/ β -catenin signaling are confined to ductal cells ^{2,3}. Since one of our goals was to maintain the organoids with cellular diversity, we developed an organoid culture protocol to bypass direct and excessive activation of EGF and WNT signaling to retain ductal and other progenitors in organoid culture. Furthermore, we observed that organoid proliferation and passaging were compromised in the absence of NRG1, NOG, RSPO1, FGFs, and TGF β (Figures 1B and S2A), indicating that multiple factors should be combined for the maintenance of salivary gland organoids.

Lastly, in the discussion section, we discussed the difference in tumor organoid media between our method and a previously reported method (Int. J. Cancer 2021; 148:193-202). A significant difference was that EGF1 and WNT3a were replaced with NRG1, as already denoted in lines 365-376 on page 17. We assume that excessive EGF signaling may inhibit the maintenance of luminal or pro-acinar cells ^{4, 5}. Our system exploited NRG1 for intermediate EGF-like signals. Furthermore, we observed c-KIT, KRT7, and AQP5 expression in tumoroids (Figures 6D and S10C), not assessed in the previous report. We referred to the reference in line 365 on page 17. We notice that all changes in revised manuscript are highlighted in yellow.

Major points:

1. In Figure S1, keratinization of organoids appeared in the presence of EGF and Wnt3a-condition medium. How much frequent keratinization happens?

Ans> We quantified the organoids' keratinization using conventional (EGF-based) and NRG1-based media. We found that keratinization of organoids increased about 9-fold (89.09%

vs. 9.88%) in the conventional medium. In addition, we measured the mRNA expression of *Involucrin*, a marker for keratinization, and added the data in Figure S1B of the revised manuscript. We observed a 4.88-fold increase in *Involucrin* mRNA in organoids cultured in the conventional medium, suggesting that NRG1-based media are more suitable for keeping epithelial features intact.

2. NGR can substitute EGF and Wnt3 through increasing endogenous Wnt expression. To confirm this possibility, it is necessary to examine whether a decrease of Wnt3a expression using siRNA affects organoid formation in the presence of NGR.

Ans> Although we tried an additional experiment using siRNA several times, it was difficult to get satisfactory results. Matrigel, which is known to be negative-charged due to heparin sulfate proteoglycans, hinders transfer of positive-charged lipofectamine-based siRNA carriers to organoids^{6,7}. Some researchers have suggested a method to increase the efficiency of siRNA⁸⁻¹⁰, and we tried them, but it did not work at all in our organoid culture system. Due to the lack of efficient siRNA delivery methods in our experiments, we alternatively tested inhibitors of the WNT/beta-catenin pathway, using IWP-2, an inhibitor of Porcn for inhibiting WNT secretion. Growth of organoids treated with IWP-2 was slower than that of untreated organoids, and there was a clear difference in cell viability (Figure S1H). These results suggested that endogenous WNT secretion is involved in salivary gland organoid formation. However, IWP-2 can inhibit not only Wnt3a but also other Wnt families. Furthermore, we observed that WNT4 and WNT5b are expressed in ductal lineages of mouse adult salivary gland, while WNT3a is expressed in embryonic tissue at scRNA-seq database (<https://sgmap.nidcr.nih.gov/sgmap/sgexp.html>). Consistently, our scRNA-seq data from human tissue and organoids supported this observation. Since WNT4 can activate both canonical- and non-canonical WNT signaling, we suggest that multiple WNT ligands have roles in maintaining salivary gland tissue homeostasis and organoid formation. A detailed mechanism associated with canonical- and non-canonical WNT signaling remains to be investigated in future studies. We added the results in Figure S1H. Please refer to lines 102-105 on page 6 and 376-383 on pages 17-18 for related details. Also, we presented some figures at the end of this response letter file.

3. In Fig.3D, calcium influx by ATP and CCh was showed. However, both dose-dependency and inhibition by antagonist atropine needed to be examined.

Ans> We tested the dose-dependency of ATP and CCh, and performed an additional experiment using atropine. ATP-induced calcium influx was observed at 10 μ M of ATP, while suboptimal at 1 μ M. We observed comparable CCh-induced calcium influx at 100 μ M and 10 μ M; however, the dose-dependency of CCh was observed between 1 and 10 μ M. We also observed that CCh-mediated calcium influx was inhibited by atropine, a muscarinic antagonist. Please refer to lines 197-199 on page 10 and Figure 3D (dose-dependency) and Figure S6H (antagonist test) for related details.

4. On page 12, lines 243-247, the authors mentioned that ACTA2 was also expressed in luminal organoids, implying that luminal progenitors may be a common ancestor of acinar and myoepithelial cells. However, the phenotype of differentiated cells after cell culture is not always recapitulating normal conditions *in vivo*, because myoepithelial cells in exocrine glands, such as salivary glands and mammary glands, are known to acquire multipotency after cell culture by dedifferentiation.

Ans> We agree that the phenotype of differentiated cells after *ex vivo* cell culture is not always recapitulating homeostatic condition *in vivo*, and myoepithelial cells in exocrine glands can obtain plasticity after cell culture. Although several reports highlighted that organoid culture could be considered a condition similar to the regeneration model after injury in which ACTA2⁺ cells could be de-differentiated from luminal progenitors¹¹⁻¹³, we realize that it remains elusive and requires further studies. We corrected the description to address all possible hypotheses. Please refer to lines 252-257 on page 12.

Reviewer #2 (Remarks to the Author):

In vitro primary salivary gland cell cultures including organoids generally do not exhibit the diversity of parenchymal cell types characteristic of the native glands, and are comprised mainly of duct-like cells that are not fully differentiated. The authors aim to create salivary organoids from adult salivary gland cells that exhibit diverse parenchymal cell types more characteristic of natural salivary glands. Using murine primary cells isolated based on previously published methods for adult submandibular salivary progenitor cell isolation, they use growth factors known to promote exocrine gland development and inhibitors used on other organoid systems to develop a Nrg1 and small molecule supplemented media that together with FGF1 and FGF7 fully supports long term submandibular salivary organoid growth and expansion that they call GEM. In contrast to the media formulations from the previous referenced work, GEM supports ductal cells without squamous metaplasia and with distinct basal and luminal marker expression as well as cells exhibiting limited expression of the acinar marker Aqp5 and the myoepithelial marker Acta2, but does not support acinar markers such as Mist1 or PAS positive mucin expression. They then develop a stepwise differentiation protocol where the GEM is replaced by a differentiation accelerating media (DAM) with key attributes including removal of Y27632 and DAPT, which promotes differentiation of organized groups of cells exhibiting expression both acinar (properly localized Aqp5, Mist1, and PAS) and myoepithelial (basally localized coexpression of Acta2 and K14) markers at the protein level as well as related transcriptional changes. Organoids show some level of neurotransmitter response similar to native glands. These data are a significant advance over previous work with adult cells demonstrating long term expansion with independent expansion and differentiation of the major parenchymal cell types, which they demonstrate is not accompanied by overt chromosomal instability. They establish murine and human parotid, submandibular, and sublingual salivary organoids that exhibit duct, acinar, and myoepithelial cell features and are reminiscent of the respective parental glands. They adapt their methods to create organoids from benign and malignant human salivary tumors, called tumoroids. Human tumor-derived tumoroids were obtained from 3 different tumor types and the tumoroids were shown to recapitulate the original tumor type. Tumoroids were then shown to be amenable to patient-specific in vitro treatment sensitivity assays using small proof of principle assays with nutlin-3 resistant growth tests or in vitro irradiation viability tests, suggestive of future clinical applicability in precision medicine

approaches. The manuscript is well written, the data are compelling, and the work is significant with potential future clinical applications.

Ans> We greatly appreciate your kind response. We performed additional experiments to address your questions, and added more detailed explanations for better understanding. We have also replied to your comments below. We notice that all changes in revised manuscript are highlighted in yellow.

There are a few points that should be clarified:

Figure 1 and S1. The authors reference Maimets et al. 2016 for the previous culture method upon which the current manuscript salivary organoid cultures are based. Of note, Maimets et al 2016 used EpCAM-high expression to positively select for these cells prior to salisphere formation and the current work methods section does not include this step. This is a significant difference that should be acknowledged in the text.

Ans> As you recommended, we addressed the differences between the culture method and media composition used by Maimets *et al.* (2016) and ours in detail in the revised Discussion section. Since EPCAM is mainly expressed in the ductal region of salivary glands, we assumed that positive selection with EPCAM marker could exclude progenitors other than ductal progenitors. As cells prepared from tissue without isolation comprise diverse types of progenitors, our organoid culture system allows recapitulating cellular diversity in salivary glands. In addition, the description of "SMG organoids based on the previous culture method" has been edited to reduce confusion. Please refer to lines 384-395 on page 18.

Figure 4 and S4. Differences in solid vs cystic morphologies of the human salivary organoids were determined to result from whether the organoids derived from basal or luminal cells, characterized by or isolated using CD49f and CD26. It seems that the organoids derived from FACS-isolated basal or luminal cells were used to form organoids and assayed for differentiation only with human GEM, and not with human DAM. This is unfortunate and it would be of great interest to see the isolated cell populations capacity for differentiation under conditions that accelerate differentiation. The authors should explain why DAM was not used or include a comparison for DAM vs GEM media as the comparison will help establish the contributions of each type of media.

Ans> According to your suggestion, we included additional data regarding the human organoids differentiated with human DAM in the revised manuscript. Please refer to Figure

S7F. Differentiated human basal organoids were more vulnerable to depletion of growth factors in the DAM, resulting in loss of cells inside the organoids when compared to organoids in the GEM (Figure 4G). We observed ACTA2 expression of punctate pattern in KRT14⁺ cells of basal organoids, supporting the idea that ACTA2⁺ myoepithelial cells are differentiated from KRT14⁺ myoepithelial progenitors. Meanwhile, MIST1 expression from AQP5⁺ cells was observed in the DAM, consistent with our organoid from non-isolated cells (Figure 3C). Interestingly, KRT5⁺KRT7⁺ cells were detected in luminal organoids in the DAM, suggesting KRT5⁺KRT7⁺ intermediate cells during luminal to basal cell transition as like as luminal progenitors in mammary gland¹¹. Please refer to lines 247-257 on page 12.

Figure 5 and S5. scRNA-seq analysis show that some of the organoid clusters are duct-like, some are acinar-like and several show incomplete differentiation with characteristics of more than one epithelial cell type. Gland specific differences are noted. Could the authors explain in more detail how they came up with the number of clusters that they report and comment on the stability of their clustering?

Ans> We appreciate the reviewers for noting the characteristics of each cluster we have generated and reported. For robust clustering, we first removed ambient RNA contamination, performed batch correction using different tools^{11, 12}, and performed graph-based clustering at multiple resolutions. Then, we applied the binary marker gene detection algorithm, assuming that different clusters should have several marker genes that robustly distinguish them with binary logic. We then selected an optimal resolution that does not lead to over-clustered segments without specific marker genes. Markers of basal and proliferating clusters, and those of other clusters, are shown in Figure S8C-S8D. Please refer to lines 798-802 on page 35 of the revised Method section for related details.

Although 8 clusters are labeled, the cell identify of each cluster is not made clear in the main figure.

Ans> While our organoids show signs of epithelial polarization and differentiation markers matching to the salivary gland tissue, some differences make it challenging to label some clusters with matching *in vivo* cell types. To minimize the potential confusion by hard-assignment of cell annotation, we would like to keep the current numerical labeling as is, and compensate for this by providing a complete list of marker genes, which has now been added to Figures S8C and S8D. In organoids, cluster 1 represented basal cell cluster of tissue, and

cluster 2 was in actively proliferating cycle (Figure S8F), supporting that basal duct cells are progenitors in the adult salivary gland, as previously reported¹⁴. Clusters 3 and 4 exhibited some gene signature of the luminal duct and acinar cells, and cluster 5 displayed some gene features observed in KLK1⁺ duct cells in salivary gland tissues. Clusters 6 and 7 seemed to be in transitional status, located between cluster 1 (basal) and differentiated clusters 3, 4, and 5 both in UMAP (Figure 5A) and pseudotime trajectory (Figure 5E). Cluster 8 had a relatively low number of cells and features of the mesenchymal cell, such as collagen genes (Figure S8D). Our original data of scRNA-seq were analyzed after 1 month of culture; however, new scRNA-seq, in which cluster 8 was not detected, was performed after 3 months of culture. Altogether, these data suggest that some mesenchymal cells survived for 1 month but were removed during long-term culture, as our culture method is prone to maintenance of epithelial cells.

ScRNA Seq has been published previously for adult SMG by Hauser, et al 2020, and they identified more clusters in adult glands in vivo. It would be helpful to know how the organoids relate to the in vivo glands. This could potentially be accomplished by direct comparison with the published data.

Ans> The data of Hauser *et al.* (2020) was obtained from the murine tissue, while our data was from the human tissue. Therefore, a direct comparison would be difficult since organoid contains differentiation characteristics, which reflect some of the aspects observed in fully differentiated and complex tissues. To our knowledge, scRNA-seq data of long-term cultured human salivary gland organoids have not yet been reported. This study focused more on the features of salivary progenitor cells with long-term culture. The role of proliferation, which is maintained by basal cells in the tissue, is reproduced in our organoid, as our data shows a distinct population of proliferating cells expressing basal cell marker. Other cell clusters, which appear to be partially differentiated epithelial progenitor cells, need further evaluation for their differentiation capacity through testing in various differentiation conditions.

Reviewer #3 (Remarks to the Author):

Yoon et al demonstrated that "Long-term salivary gland organoid culture maintains distinct glandular properties of murine and human major salivary glands". Authors suggested that a modified culture method for the generation of salivary glands organoid was suitable to long-term culture, and maintain the major salivary glands cell types in human and murine organoids. And the authors also presented culture conditions for tumor salivary gland organoids. Their overall findings are interesting, however, there are several issues in the manuscript and the data as presented, which should be addressed as follows.

In this study, the authors emphasized that salivary gland organoids can be cultured for a long time and retain their properties. However, I could not see the comparing controls to describe the characteristics of the long-term culture. And I also could not find the time-dependent experiment, such as day or month, for assessment of long-term culture properties. And also I could not find the quantitative analysis (ex, qRT-PCR analysis) in the main figures.

Ans> According to your recommendations, we performed additional experiments to validate our hypothesis that salivary gland organoids can be maintained for a long-term period. We also added quantitative analysis for gene expression in our revised manuscript, which was consistent with our imaging and RNAseq results. We notice that all changes in revised manuscript are highlighted in yellow.

1. To make the GEM, the authors used NRG1 to replace the Wnt3A condition medium. Because NRG1 treatment raises the expression of various Wnt genes at the endogenous level, the authors assured that NRG1 could be substituted with the addition of Wnt 3a. Is the endogenous levels of Wnts expression sufficient to produce salivary gland organoids? The expression of not only Wnt3a but also Wnt4, Wnt5b, and Wnt10b is increased 10-fold or more by treatment with NRG1. In order to confirm that the endogenous level of Wnt3a concentration is sufficient, downregulation of Wnt3a after NRG1 treatment will be required, and also confirmation of other Wnt parts will be required during the generation of organoids.

Ans> Although we tried additional experiments as you recommended, it was impossible to down-regulate gene expression in organoids cultured in Matrigel using lipofectamine-siRNA complex. The lipofectamine-siRNA complex cannot penetrate Matrigel, since Matrigel is strongly negative-charged and Lipofectamine is positive-charged.

Instead, we used IWP2, an inhibitor of Porcn, to inhibit Wnt secretion. Growth of organoids

treated with IWP-2 was slower than that of untreated organoids. These results suggested that the Wnt pathway is involved in salivary gland organoid formation. However, those reagents could inhibit not only Wnt3a, but also other Wnt families. Furthermore, we observed that WNT4 and WNT5b are expressed in the ductal lineages of adult mouse salivary gland, while WNT3a is expressed in embryonic tissue in the scRNA-seq database (<https://sgmap.nidcr.nih.gov/sgmap/sgexp.html>). Consistently, our scRNA-seq data from human tissue and organoids supported this observation. Since WNT4 can activate both canonical- and non-canonical WNT signaling, we suggest that multiple WNT ligands have roles in maintaining salivary gland tissue homeostasis and organoid formation. A detailed mechanism associated with canonical- and non-canonical WNT signaling should be investigated in further studies. We added the results in Figure S1H. Please refer to lines 102-105 on page 6 and 376-383 on pages 17-18 for related details. We also presented some figures at the end of this response letter file.

2. In figure 1C, D, E. How many days is 30 passage? The authors showed 9 days organoid (mouse) or 12 days organoid (human) in the main figures. Authors should present the change of cell type marker expression (qRT-PCR analysis) and phenotypes by date.

Ans> First, mouse submandibular gland organoids at passage 30 corresponded to culture for 7 months. Next, we measured the expressions of *Krt5*, *Krt7*, *Krt14*, *Aqp5*, *Acta2*, and *Bhlha15* genes in mouse SMG tissues and organoids cultured for 1, 4, and 8 months. Although gene expressions in organoids were different from SMG tissues (upregulated; *Krt5*, *Krt7*, *Krt14*, and *Aqp5*, downregulated; *Bhlha15* and *Acta2*), these observations were consistent with the results of our bulk RNA-seq and previous reports¹⁵. Regarding to long-term maintenance of gene expression, we observed sustained gene expressions of several markers, including *Krt5*, *Krt7*, *Aqp5*, and *Krt14*, throughout the culture period. Since limited potency of long-term culture was observed in the expressions of *bhlha15* and *Acta2* (Figure S2C), we speculate that acinar- or myoepithelial-specific optimized culture will allow long-term cultivation of these cells, which is now under investigation. We added the results in Figure S2C. Please refer to lines 109-111 on page 6 for related details.

3. Figure 1F and G, authors should perform re-analysis using GEM and DAM RNA seq result compared to mouse tissue for assessment of organoid in GEM and DAM condition.

Ans> We re-analyzed RNA-seq with data from the murine tissues, and compared the

organoids in the GEM and DAM conditions. Figures 1F and 1G were replaced with figures with new data. We observed enriched expression of genes associated with ductal lineage (*Krt5*, *Krt7*, *Krt14*, *Krt19*, *Epcam*, *Cdh1*, *Tjp1*, and *Vipr1*), proliferation (*Pcna*, *Mki67*, *Myc*, and *Cdkn1a*), *Aqp5*, and *Trp63*, while the expression of genes related to differentiated acinar (*Chrm3*, *Smgc*, and *bhlha15*) or myoepithelial cells (*Acta2*) were reduced compared to the SMG tissue (Figure 1F), which was also observed in the human organoid data (Figures S5C). Furthermore, we revised Figure 1G to show differentially expressed genes with tissue results. However, these data must be interpreted with caution, as the RNAseq results from tissue might include gene expression from non-epithelial cells, such as endothelial, immune, or stromal cells. Please refer to lines 128-131 on page 7.

4. Figure 2C and E,F,G. Authors should perform qRT-PCR analysis for validation of each expression level. Only ICC data and RNA-seq data are not enough for elucidation of the result.

Ans> For Figure 2C, we conducted qRT-PCR for *Muc19*, and *Amy2*. As shown in the IF, *Amy2* transcript was expressed about 4.5 times more in mPG organoids than in mSLG organoids, and *Muc19* transcript was expressed about 124.5 times more in mSLG organoids than in mPG organoids. We added the results to Figure 2D, and explained them in lines 155 on page 8.

For Figures 2F-H, *Calml3*, *Serpine2*, *Sox2*, *Dcpp3*, and *Itga8* were verified by qRT-PCR, and the results were confirmed to be the same as those of volcano plots of bulk mRNA sequencing. In addition, these qRT-PCR results showed a similar tendency to the published database (<https://sgmap.nidcr.nih.gov/sgmap/sgexp.html>). We propose *Calml3* and *Serpine2* as PG-, *Itga8* as SMG-, and *Sox2* and *Dcpp3* as SLG-specific markers validated in RNA-seq and qRT-PCR. We added the results to Figure S4, and explained them in lines 166-168 on page 8.

5. Figure 3. Authors also should present the change of marker expression (qRT-PCR analysis) and phenotypes by date. In this system, the author demonstrated the 3 month culture of organoids. These are important points in this study. After long-term culture, the cell types and specific cell type markers should be maintained. Thus authors should present the qRT PCR results of cell type markers by date or month. Moreover in figure 3D, the authors suggest the result of organoids function by neurotransmitters (ATP, CCh). The Authors also should

exhibit the result of calcium influx by date or month. Summarizing these results, I can decide that the new culture system is better than the old system for long-term culture.

Ans> We performed qRT-PCR to assess the gene expression of cultured organoid cell type markers for 1, 2, and 4 months. Gene expression patterns of ductal, acinar, and myoepithelial cell markers appeared to be consistent over time. Although acinar markers and ACTA2 were reduced compared to tissue expression, these phenotypes were not result from long-term culture. The enrichment of ductal lineage in human organoids was consistent with our system's RNA-seq analysis with murine SMG organoids. Please refer to lines 189-191 on page 9 and Figures S5C for related details. In addition, we performed calcium influx experiments with human SMG organoids maintained for 3 months. The data showed that our long-term organoid culture sustained its functionality. Original figures were transferred to Figure S6E (1-month culture), and new results are displayed in Figure 3D (3-month culture). Please refer to lines 197-199 on page 10 for related details.

6. Figure 5A. It is not clear for performing scRNA-seq in this study. According to the tile, it is assumed that this study established "salivary gland organoid long-term culture." In general, for scRNA-seq data, differentiation of salivary gland organoids needs to show how they have corresponded to the cell types. However, this does not mean the results of long-term maintenance of organoids. This study needs to provide the data to demonstrate the purpose of long-term maintenance.

Ans> For validation of long-term maintenance as you recommended, we performed scRNA-seq on long-term cultured organoid samples and compared the data with the original single-cell data. After processing the long-term organoid single-cell data, it was merged with the original data and corrected for batch-effect using HARMONY. In the UMAP projection of the combined data, the long-term cultured data overlapped well with the original data while maintaining the structure of original clusters, except cluster 8 in which some mesenchymal features were observed (Figure S8D). Since our human GEM contained Noggin and A83-01, TGF- β pathway inhibitors and cluster 8 were relatively small, we speculated that a few mesenchymal cells remained during short-term culture (1 month in Figure 5), but were removed, and epithelial features were reinforced in long-term culture (3 months in Figure S9). We noticed that our new data had different shapes of UMAP, since scRNA-seq of long-term cultured organoid was performed with only SMG organoids, and combined UMAP were illustrated with SMG organoids cultured for 1 month (early passage) and 3 months (late

passage) (Figure S9A). If clustering is performed with new UMAP, this will create clusters and cluster-specific gene signatures, which is quite different from our original data. Therefore, we analyzed our new scRNA-seq results using the original figures' clustering strategy. Altogether, these data indicate that the robustness in organoid cell clustering is well-preserved after long-term culture. Please refer to Figure S9 and lines 296-303 on page 14 for related details.

7. scRNA-seq : This study should compare the well-differentiated salivary gland organoid with long-term maintenance salivary gland organoid in cell-type differences in order to highlight the importance of this protocol. From this data, there is no clue for demonstrating long-term salivary gland organoids are well maintained. Actual salivary gland organoid data of long-term maintenance should be provided.

Ans> As we explained in our response to question #6, long-term cultured organoids were well-maintained in the scRNA-seq data. Also, the data showed that basal cells (cluster #1) and proliferating cells (cluster #2), which are essential for organoid maintenance, were well-conserved after long-term culture. Altogether, these data indicate that our organoid preserved its function and characteristics after long-term culture.

6. Figure 6. Authors also should present the expression level of cancer markers (qRT-PCR analysis) compared to normal organoid and tissue.

Ans> According to your comment, we performed qRT-PCR to assess tumor-specific gene expression with tissue and organoids cultured in tumor-GEM and maintained for at least 1 month. Our data demonstrated that differential gene expressions of *PLAG1* (for pleomorphic adenoma), *KIT* (for adenoid cystic carcinoma), or *MUC1* (for mucoepidermoid carcinoma) were conserved both in tissue and organoids. Although the expression of *PLAG1* was reduced in overall organoid culture, as *PLAG1* is mainly expressed in myoepithelial cells of pleomorphic adenoma, these data consistently showed that our organoid system is enriched with ductal compartment, while other acinar of myoepithelial lineages are also maintained. We added new data to Figure S10E. Please refer to lines 331-333 on page 15 for related details.

7. Describe the figure legends in more detail.

Ans> We revised our figure legends with more details, as you suggested.

Figure 1. Nomenclature of cell cluster of adult mouse salivary gland defined in open database of NIDCR

Figure 2. *Wnt4* expression in adult mouse salivary gland

Figure 3. *Wnt5b* expression in adult mouse salivary gland

Figure 4. *WNT3A*, *WNT4*, and *WNT5B* expression in human salivary gland tissues

Figure 5. *WNT3A*, *WNT4*, and *WNT5B* expression in human salivary gland organoids

References

1. Chen G, Gulbranson DR, Yu P, et al. Thermal stability of fibroblast growth factor protein is a determinant factor in regulating self-renewal, differentiation, and reprogramming in human pluripotent stem cells. *Stem Cells* 2012; 30: 623-630. 2012/01/04. DOI: 10.1002/stem.1021.
2. Miyoshi K, Rosner A, Nozawa M, et al. Activation of different Wnt/beta-catenin signaling components in mammary epithelium induces transdifferentiation and the formation of pilar tumors. *Oncogene* 2002; 21: 5548-5556. 2002/08/08. DOI: 10.1038/sj.onc.1205686.
3. Zuo WL, Yang J, Gomi K, et al. EGF-Amphiregulin Interplay in Airway Stem/Progenitor Cells Links the Pathogenesis of Smoking-Induced Lesions in the Human Airway Epithelium. *Stem Cells* 2017; 35: 824-837. 2016/10/07. DOI: 10.1002/stem.2512.
4. Karthaus WR, Iaquinta PJ, Drost J, et al. Identification of multipotent luminal progenitor cells in human prostate organoid cultures. *Cell* 2014; 159: 163-175. 2014/09/10. DOI: 10.1016/j.cell.2014.08.017.
5. Hosseini ZF, Nelson DA, Moskwa N, et al. Generating Embryonic Salivary Gland Organoids. *Curr Protoc Cell Biol* 2019; 83: e76. 2018/11/06. DOI: 10.1002/cpcb.76.
6. Park H, Kim M, Kim HJ, et al. Heparan sulfate proteoglycans (HSPGs) and chondroitin sulfate proteoglycans (CSPGs) function as endocytic receptors for an internalizing anti-nucleic acid antibody. *Sci Rep* 2017; 7: 14373. 2017/11/01. DOI: 10.1038/s41598-017-14793-z.
7. Zuris JA, Thompson DB, Shu Y, et al. Cationic lipid-mediated delivery of proteins enables efficient protein-based genome editing in vitro and in vivo. *Nat Biotechnol* 2015; 33: 73-80. 2014/10/31. DOI: 10.1038/nbt.3081.
8. Morgan RG, Chambers AC, Legge DN, et al. Optimized delivery of siRNA into 3D tumor spheroid cultures in situ. *Sci Rep* 2018; 8: 7952. 2018/05/23. DOI: 10.1038/s41598-018-26253-3.
9. Zhang Q, Pan Y, Yan R, et al. Commensal bacteria direct selective cargo sorting to promote symbiosis. *Nat Immunol* 2015; 16: 918-926. 2015/08/04. DOI: 10.1038/ni.3233.
10. Zoldan J, Lytton-Jean AK, Karagiannis ED, et al. Directing human embryonic stem cell differentiation by non-viral delivery of siRNA in 3D culture. *Biomaterials* 2011; 32: 7793-7800. 2011/08/13. DOI: 10.1016/j.biomaterials.2011.06.057.
11. Guo W, Keckesova Z, Donaher JL, et al. Slug and Sox9 cooperatively determine the mammary stem cell state. *Cell* 2012; 148: 1015-1028. 2012/03/06. DOI: 10.1016/j.cell.2012.02.008.
12. Date S and Sato T. Mini-gut organoids: reconstitution of the stem cell niche. *Annu Rev Cell Dev Biol* 2015; 31: 269-289. 2015/10/06. DOI: 10.1146/annurev-cellbio-100814-125218.
13. Jarde T, Chan WH, Rossello FJ, et al. Mesenchymal Niche-Derived Neuregulin-1 Drives Intestinal Stem Cell Proliferation and Regeneration of Damaged Epithelium. *Cell Stem Cell* 2020; 27: 646-662 e647. 2020/07/22. DOI: 10.1016/j.stem.2020.06.021.
14. Maimets M, Rocchi C, Bron R, et al. Long-Term In Vitro Expansion of Salivary Gland Stem Cells Driven by Wnt Signals. *Stem Cell Reports* 2016; 6: 150-162. 2016/01/05. DOI: 10.1016/j.stemcr.2015.11.009.

15. Shubin AD, Sharipol A, Felong TJ, et al. Stress or injury induces cellular plasticity in salivary gland acinar cells. *Cell Tissue Res* 2020; 380: 487-497. 2020/01/05. DOI: 10.1007/s00441-019-03157-w.

REVIEWERS' COMMENTS

Reviewer #1 (Remarks to the Author):

The authors appropriately responded to the requests raised by me. So, I feel that the revised manuscript has been improved by the authors.

Reviewer #2 (Remarks to the Author):

The authors have sufficiently answered my queries and addressed my comments.

Reviewer #3 (Remarks to the Author):

Reviewer#3

I appreciate the authors' responses to my concerns. And The authors have addressed my comments and the resulting changes to the manuscript